# A new automatic approach for extracting glacier centerlines

Dahong Zhang[1,2], Xiaojun Yao[1,3], Hongyu Duan[1], Shiyin Liu[4], Wanqin Guo[5], Meiping Sun[1], Dazhi Li[1]

[1]College of Geography and Environment Sciences, Northwest Normal University, Lanzhou, China

[2]College of Urban and Environmental Sciences, Northwest University, Xi'an, China

[3]National Cryosphere Desert Data Center, Lanzhou, China

[4]Institute of International Rivers and Eco-security, Yunnan University, Kunming, China

[5]State Key Laboratory of Cryospheric Science, Northwest Institute of Eco-Environment and Resources, Chinese Academy of Sciences, Lanzhou, China

*Correspondence to*: Xiaojun Yao (yaoxj_nwnu@163.com)

**Abstract.** Glacier centerlines are crucial input for many glaciological applications. From the morphological perspective, we proposed a new automatic method to derive glacier centerlines, which is based on the Euclidean allocation and the terrain characteristics of glacier surface. In the algorithm, all glaciers are logically classified as three types including simple glacier, simple compound glacier and complex glacier, with corresponding process ranges from simple to complex. The process for extracting centerlines of glaciers introduces auxiliary reference lines, and follows the setting of not passing through bare rock.

The program of automatic extraction of glacier centerlines was implemented in Python and only required glacier boundary and digital elevation model (DEM) as input. Application of this method to 48571 glaciers in the second Chinese glacier inventory automatically yielded the corresponding glacier centerlines with an average computing time of 20.96 s, a success rate of 100% and a comprehensive accuracy of 94.34%. A comparison of the longest length of glaciers to the corresponding glaciers in the Randolph Glacier Inventory v6.0 revealed that our results were more superior. Meanwhile, our final product provided more

information about glacier length, such as the average length, the largest length, the lengths in the accumulation and ablation regions of each glacier.

## 1 Introduction

Glacier is an important freshwater resource on earth and a vital part of the cryosphere (Muhuri et al., 2015). According to the Fifth Assessment Report (AR5, https://www.ipcc.ch/) published by the Intergovernmental Panel on Climate Change (IPCC),

there are 168331 glaciers (including ice caps) in the world, with a total area of 726258 km$^2$ apart from ice sheets. Glaciers

move towards lower altitude by gravity, which is the most obvious distinction between glacier and other natural ice bodies.

The glacier flowlines are the motion trajectories of a glacier, and the main flowline is the longest flow trajectory of glacier ice.

Due to the differences in the speed and moving direction of any point at the surface or inside the glacier, the calculation of the

main flowline of glaciers requires a coherent velocity field data, which is difficult to obtain on the global or regional scale

(McNabb et al., 2017). Alternatively, some concepts such as the glacier axis and the glacier centerlines were proposed (Le Bris

and Paul, 2013;Machguth and Huss, 2014;Kienholz et al., 2014).

As an important model parameter, glacier centerline can be used to determine the change of glacier length (Leclercq et al.,

2012a;Nuth et al., 2013), analyze the velocity field (Heid and Kääb, 2012;Melkonian et al., 2017), estimate the glacier ice

volume (Li et al., 2012;Linsbauer et al., 2012), and develop one-dimensional glacier model (Oerlemans, 1997;Sugiyama et al.,

2007). Meanwhile, the length of the longest glacier centerline is one of the key determinants of glacier geometry and an

important parameter of glacier inventory (Leclercq et al., 2012b;Paul et al., 2009). The length and area of glacier can be also

used to estimate the large-scale glacier ice volume (Zhang and Han, 2016;Gao et al., 2018). The length change at the terminus

of a glacier can directly reflect the state of motion, e.g., glacier recession, glacier advance or surging (Gao et al., 2019).

Winsvold et al. (2014) analyzed the changes of glacier area and length in Norway, using glacier inventories derived from

Landsat TM/ETM+ images and digital topographic maps. Herla et al. (2017) explored the relationship between the geometry

and length of glaciers in the Austrian Alps based on a third-order linear glacier length model. Leclercq et al. (2012)

reconstructed annual averaged surface temperatures in the past 400 years on hemispherical and global scale from glacier length

fluctuations (Leclercq et al., 2014). These studies indicated that both the extraction of contemporary glacier length and the

reconstruction of historical glacier length require more accurate automatic extraction methods of glacier flowlines.

In order to obtain the length of glaciers, some automatic or semi-automatic methods were proposed in recent years. Schiefer

et al. (2008) extracted the longest flow path on the ice surface based on a hydrological model, which was generally 10% to

15% larger than the glacier length. Le Bris et al. (2013) accomplished the automatic extraction of flow lines from the highest

point to the terminus of a glacier based on the concept of glacier axis, with a verification accuracy of 85%. Unfortunately, the

branches of glacier centerlines have not been extracted and the length is not necessarily the maximum for huge or complex

glaciers (Paul et al., 2009). Machguth et al. (2014) proposed an extraction method of glacier length based on the slope and

width of glacier with a success rate of 95-98%, however the branches of glacier centerlines could not be extracted either.

Kienholzs et al. (2014) applied the grid–least-cost route approach to the automatic extraction of glacier flow lines, having an

automation degree of 87.8% with additional manual intervention. Yao et al. (2015) proposed the semi-automatic method of

extraction glacier centerlines based on Euclidean allocation theory, which required the expertise and experiences for composite

valley glaciers and ice caps. The aims of this study are to design an algorithm to: (i) automatically generate centerlines for the

main body of each glacier and its branches; (ii) automatically calculate the longest length, average length, the length of

accumulation region, and the length of ablation region of each glacier, along with corresponding polylines; and (iii) improve

the degree of automation as much as possible on the premise of ensuring the accuracy of glacier centerlines.

## 2 Input data and test region

The glacier dataset used in this study is the Second Chinese Glacier Inventory (SCGI) released by National Tibetan Plateau

Data Center (http://westdc.westgis.ac.cn/data), which has been approved by some organizations (e.g., WGMS, GLIMS,

NSIDC, etc.) and adopted in the Randolph Glacier Inventory (RGI) v6.0 (Guo et al., 2017). According to the SCGI (Fig.1),

there were 48571 glaciers in China, with a total area of 51766.08 km$^2$, accounting for 7.1% of the glacier area in the world

except for the Antarctic and Greenland ice sheets (Liu et al., 2015). Due to the lack of automatic method to calculate glacier's

length, there was no length property in the SCGI, and some subsequent studies haven't made great breakthroughs (Yang et al.,

2016;Ji et al., 2017).

The SCGI was produced based on Landsat TM/ETM+ images and ASTER images in the period of 2004-2011 and SRTM v4.1

with a spatial resolution of 90 m (Liu et al., 2015). In this study, we selected SRTM1 DEM v3.0 (http://www2.jpl.nasa.gov/srtm,

last accessed on March 2, 2013, with a spatial resolution of 30 m) (Farr et al., 2007) in consideration of its free access and

higher data quality, which was used to identify division points on the glacier outlines, extract ridge lines in the coverage region

of glaciers, and generate the glacier centerlines. Additionally, we extracted glacier data in China from the RGI v6.0 provided

by GLIMS (http://www.glims.org/RGI/). There are 38053 glaciers matching the graphic position of the SCGI. The field of $L_{max}$

of RGI v6.0 provides the length of the longest flowlines on the glacier surface, which was calculated with the algorithm

proposed by Machguth et al. (2014). For verifying the validity and accuracy of glacier centerlines, we compared the extracted

longest length of glaciers with the value of $L_{max}$ in the RGI v6.0.

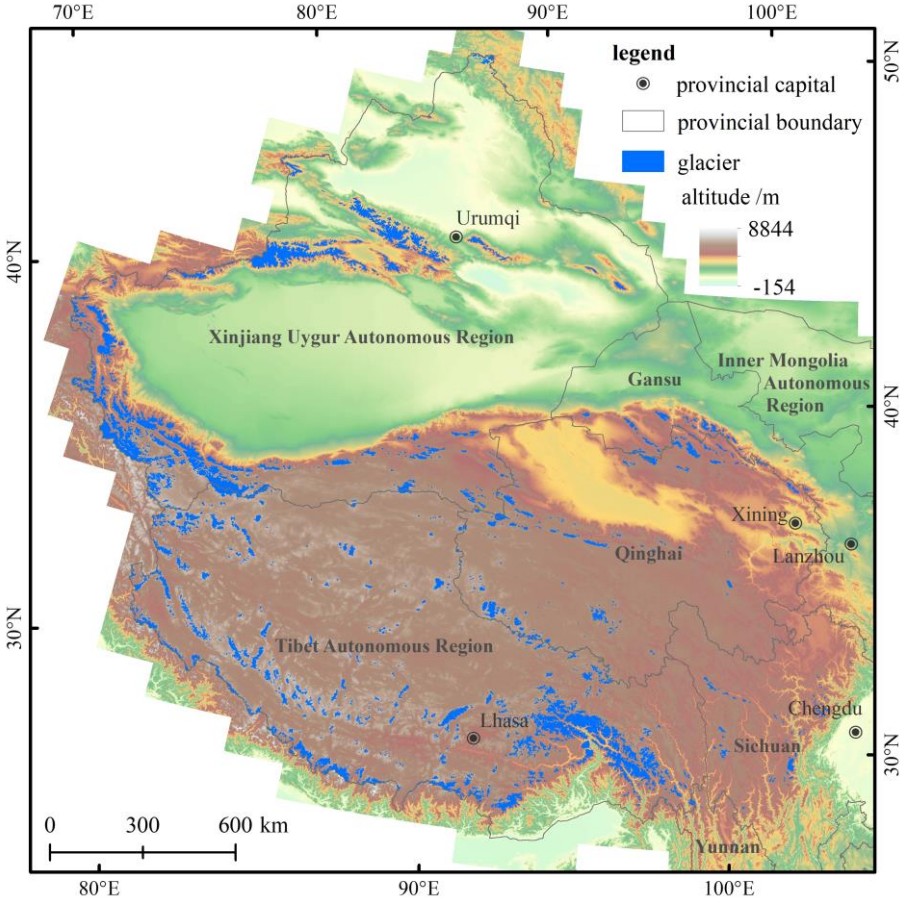

**Figure 1: The distribution of glaciers in China.**

## 3 Principles and algorithm of glacier centerline extraction

In order to implement the automatic extraction of glacier centerlines, we have designed a new set of algorithms. Relevant

parameters and processing procedures are introduced as follows.





### 3.1 Model parameters

The code was written in Python and partially invoked the site package of arcpy. The calculation of the glacier centerlines relies

on two basic inputs: (i) glacier in the form of polygon with a unique value field and a projection coordinate system (unit: m),

(ii) DEM data having the spatial resolution and acquisition time close to the images used for glacier inventory. We defined 11

adjustable parameters named $P_i$ ($i$=1,…,11) (see Table 1), which were achieved by classifying glacier polygon through a set of

reasonable rules. The purpose is to improve the degree of automation and the accuracy. Three key parameters are described as:

—$P_3$: the threshold of flow accumulation, to control the generation of auxiliary lines.

—$P_6$: the step size of searching the local highest points, to control the extraction of extremely high points.

—$P_8$: The grid cell size of Euclidean allocation, to improve the algorithm efficiency.

In the algorithm, the number of the local highest points is affected by the perimeter of the glacier ($P_G$). We took the given area

($A$) and the perimeter ($P$) of the equilateral triangle corresponding to $A$ as the grading threshold (Eq.1). According to the area

($A_G$) and the perimeter ($P_G$) of each glacier, all glaciers were divided into five levels (Eq.2), which represented the five levels

of glacier polygon with difference in $P_G$. The built-in parameters were set according to the different levels (Table 1). $P_4$, $P_5$

and $P_9$ were controlled in proportion to the side length of the equilateral triangle corresponding to $P$. The proportional

coefficient was $T$ (Eq.3). According to the actual situation of the repeated programing test, the empirical value of each

parameter was given in Table 1.

$$P(A) = 2 \times 3^{0.25} \times A^{0.5} \tag{1}$$

$$L(A_G, i) = \begin{cases} i: A_G \in [A_{i-1}, A_i) \text{ and } P_G \in [P(A_i), +\infty) \text{ and } i \in (1,5] \\ i: A_G \in [A_i, A_{i+1}) \text{ and } P_G \in [P(A_i), P(A_{i+1})) \text{ and } i \in [1,5] \\ i: A_G \in [A_{i+1}, A_{i+2}) \text{ and } P_G \in (0, P(A_{i+1})) \text{ and } i \in [1,5) \\ 0: \text{ the above conditions aren't met} \end{cases} : \begin{array}{l} A = \{0,1,5,20,50,+\infty\} \\ i = \{1,2,3,4,5\} \end{array} \tag{2}$$

$$f(T) = \frac{P_G}{3 \times 2 \times T} \tag{3}$$




**Table 1 The description of adjustable parameters.**

| Levels | 1 | 2 | 3 | 4 | 5 | Parameter elucidation |
|---|---|---|---|---|---|---|
| Par. | L($A_G$,1) | L($A_G$,2) | L($A_G$,3) | L($A_G$,4) | L($A_G$,5) | |
| *$P_1$ | | | "10 meters" | | | Maximum distance between adjacent vertexes of polyline |
| *$P_2$ | | | "30 meters" | | | Buffer distance outside the glacier outline |
| $P_3$ | 500 | 600 | 700 | 800 | 800 | The threshold of accumulative flow |
| $P_4$ | $f(10)$ | $f(11)$ | $f(12)$ | $f(13)$ | $f(15)$ | The length of the shortest auxiliary line |
| $P_5$ | $f(2)$ | $f(3)$ | $f(4)$ | $f(5)$ | $f(6)$ | The length of the longest auxiliary line |
| $P_6$ | 50 | 60 | 70 | 80 | 80 | The interval for searching the local highest points |
| $P_7$ | 0.2 | 0.2 | 0.5 | 0.5 | 1 | The matching tolerance of the vertexes of polyline |
| $P_8$ | 1 | 5 | 15 | 15 | 30 | The size of grid cell in Euclidean allocation |
| $P_9$ | $f(15)$ | $f(30)$ | $f(60)$ | $f(120)$ | $f(240)$ | Minimum distance between the adjacent local highest points |
| $P_{10}$ | 5 | 10 | 15 | 20 | 30 | The smoothing tolerance of polylines |
| *$P_{11}$ | | | P(A=5) | | | Threshold to control the length of the longest auxiliary line |

Note: the parameters with "*" are constant.

## 3.2 Computation flow

In this paper, glaciers were divided into three categories: simple glacier (extremely high point: single, auxiliary line: no, the area of bare rock: no), simple compound glacier (extremely high point: several, auxiliary line: no, the area of bare rock: no), and complex glacier (extremely high point: several, auxiliary line: yes, the area of bare rock: yes). Following the principle from simple to complex, the algorithm was composed of six main steps: data preprocessing, extraction of auxiliary lines, identification of division points, reconstruction of feature lines, extraction of centerlines and the calculation of glacier length. The flow chart of the algorithm is illustrated in Fig.2.

The automatic extraction of glacier centerlines obeys the following rules: (i) the elevation of the local highest points must be higher than the equilibrium line altitude (ELA), (ii) a glacier has only one exit, which is the lowest point of the outer boundary of the glacier outline ($G_{po}$); (iii) the auxiliary line only acts on the accumulation region of glacier; (iv) the $G_{po}$, auxiliary lines, and bare rock region are also used to restrict the flow direction of the glacier centerlines.



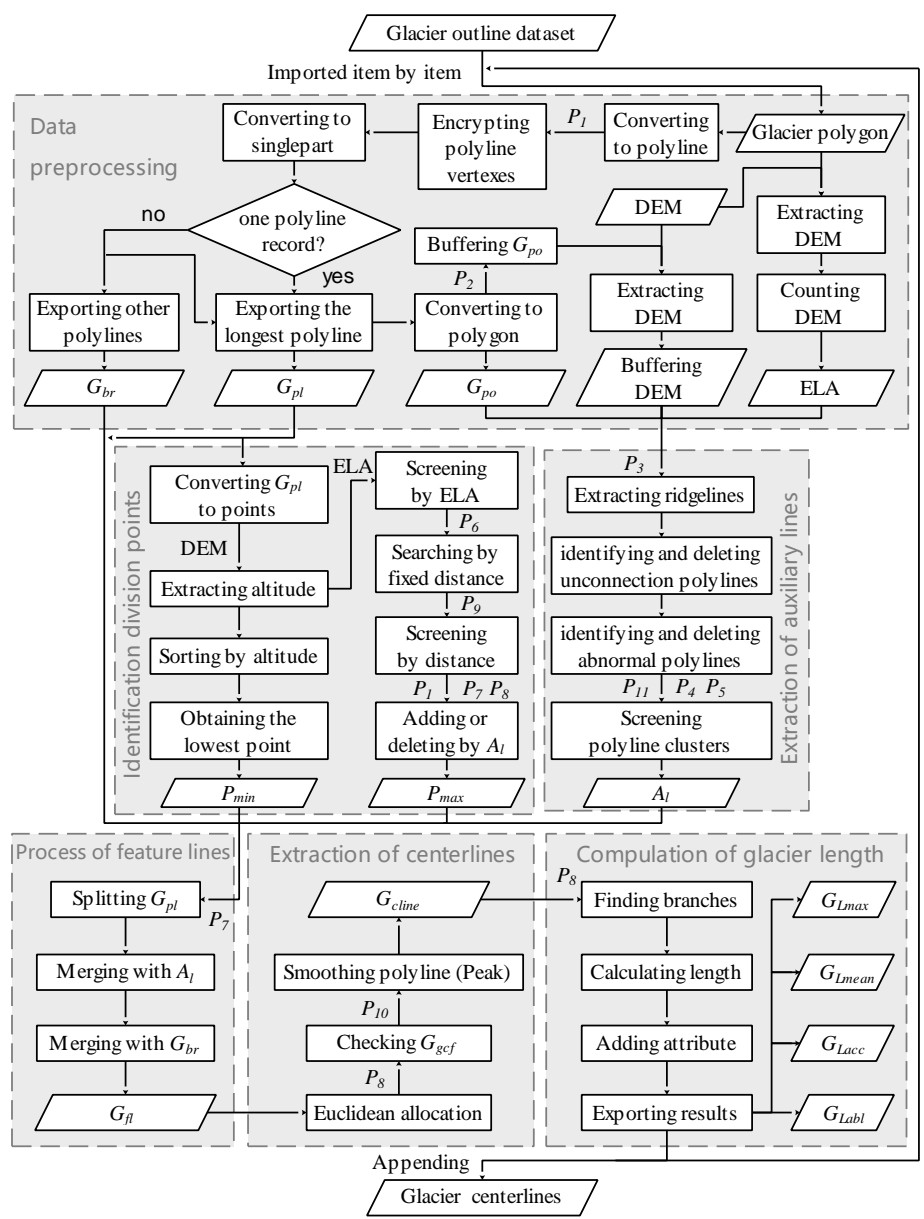

**Figure 2: The flow chart of algorithm.**

**3.3 Critical processes**

**3.3.1 Data preprocessing**



The data preprocessing includes four parts: (i) checking the input data, (ii) pre-processing the glacier outlines, (iii) fine-tuning the built-in parameters, and (iv) calculating the ELA of glaciers. First, the polyline of the outer boundary of the glacier ($G_{pl}$),

the polygon of the $G_{po}$, the boundary of the bare rock in glacier ($G_{br}$) were obtained by splitting the glacier outlines in the importing module. These temporary data would be used as the input parameters of other modules in subsequent process. Secondly, the module exported the number of closed lines in glacier outlines, $A_G$ and $P_G$, which were used to determine the number of bare rocks on the glacier surface, the type and level of glaciers. Thirdly, according to the parameter adjusting rules at the level of glaciers, 11 built-in parameters were fine-tuned. Finally, the median elevation ($Z_{min}$) of each glacier aided by its

DEM was computed, which was then used to estimate the ELA of each glacier. The schematic diagram of processing glacier outlines is shown in Fig.3.

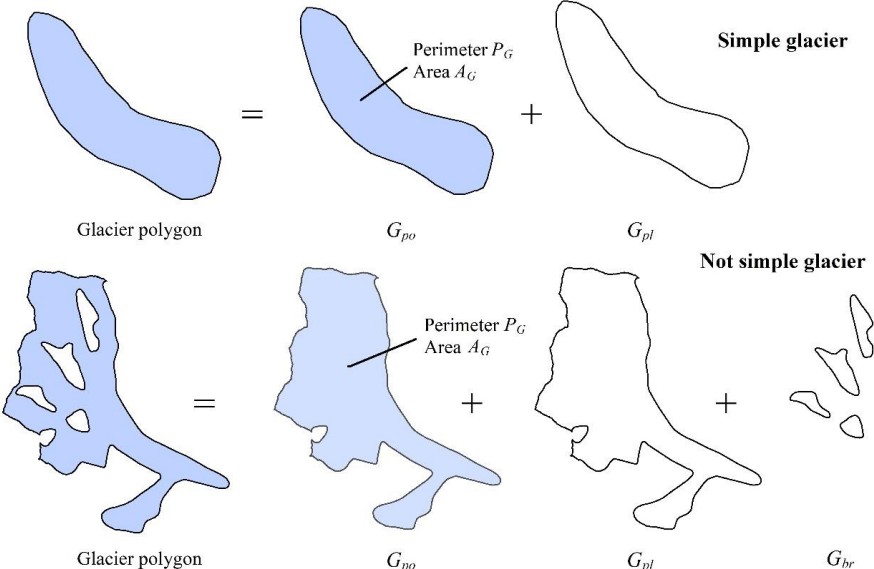

**Figure 3: The schematic of processing raw data.**

### 3.3.2 Extraction of auxiliary lines

For making glacier centerlines more reasonable, we introduced the auxiliary lines that represent the internal ridgelines of glaciers to intervene in the generation of centerline for the upper part of a glacier. The extraction of auxiliary lines included the extraction of ridgelines and post-processing. The extraction of ridgelines was easily accomplished by hydrologic analysis.




The post-processing was relatively complicated. The main reason was that the auxiliary lines were tree-like polylines starting from the upper boundary of the glacier. In principle, the material flow in the location of the auxiliary lines on the glacier surface could be obviously blocking-up, which was equivalent to the ice divide. The preliminary ridgelines needed to be screened once more combining with DEM by traversal method. Determining the cluster of auxiliary lines was the main problem to be solved by the algorithm of this part. According to the designed algorithm, it could be divided into five parts in post-processing: (i) identifying and deleting the disconnected lines, (ii) identifying and deleting the abnormal lines, (iii) determining the members of line cluster, (iv) determining the longest length of line cluster, and (v) screening the line clusters. The schematic diagram of extracting the auxiliary lines is shown in Fig.4.

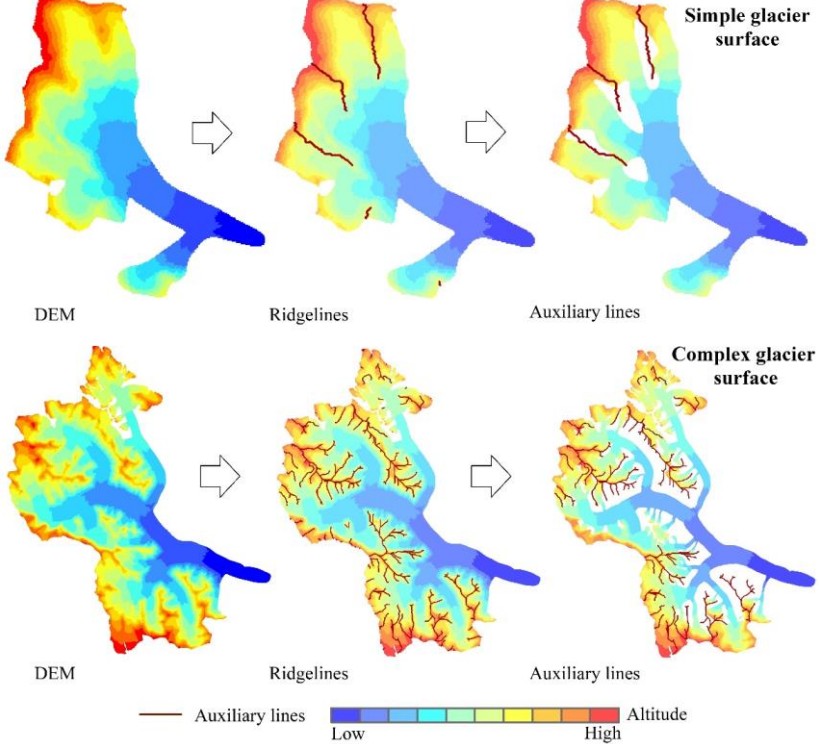

**Figure 4: The schematic of extracting auxiliary lines.**

The automatically extracted ridgelines were often disconnected, and it was necessary to remove independent existence or unreasonable ridgelines using the auxiliary data such as DEM, ELA and $G_{po}$ by ergodic algorithms. Firstly, the ridgelines of the glacier surface ($A_s$) were obtained by clipping the ridge lines using $G_{po}$. The set of all possible starting points of auxiliary



lines was gained by intersecting $A_s$ with $G_{pl}$. Then, the ridgeline clusters connected to each starting point were achieved and marked by traversing the point set. The number of auxiliary lines was initially determined. Lastly, the longest length of each auxiliary line was calculated by adopting the critical path algorithm. The final auxiliary lines ($A_l$) were obtained by screening all auxiliary lines using the three parameters of $P_4$, $P_5$ and $P_{11}$.

### 3.3.3 Identification of division points

The division points include the lowest point ($P_{min}$) and the local highest point ($P_{max}$). The ordered point set ($h$) was obtained after converting $G_{pl}$ from a polyline to a point set and extracting the elevation for the point set. The method for obtaining $P_{min}$ was relatively simple, as showed in Eq. (4).

$$P_{min} = Min(h_1, h_2, \cdots, h_n) \tag{4}$$

In comparison, the extraction of $P_{max}$ was more complicated. It was necessary to ensure the extraction of all possible branches of the centerlines and avoid the redundancy of branches. The algorithm could be divided into four steps: (i) obtaining the local highest point set (M″) by filtering $h$ (Eq.5, Eq.6) according to $P_6$, (ii) removing the elements (Eq.7) at an altitude lower than ELA from M″, (iii) removing the elements (Eq.8) of adjacent distance less than $P_9$ from M′, and (vi) checking, deleting or adding some local highest points (Eq.9) using the auxiliary lines to ensure that there was at least one local highest point among adjacent auxiliary lines.

$$H_i = \left\{ h_{i-\frac{P_6}{2}}, \cdots, h_{i-1}, h_i, h_{i+1}, \cdots, h_{i+\frac{P_6}{2}} \right\}, i \in [\tfrac{P_6}{2}, n - \tfrac{P_6}{2}] \tag{5}$$

$$M'' = \{h_i | h_i \geq Max(H_i)\} \tag{6}$$

$$M' = \{M''_j | M''_j \geq ELA\}, j \in [0, card(M'')) \tag{7}$$

$$M = \{M'_k | d(M'_{k-1}, M'_k) \geq P_9, and\ d(M'_k, M'_{k+1}) \geq P_9\}, k \in [0, card(M')) \tag{8}$$

$$P_{max} = M \cup \{l_j | l_j \geq Max(L_i)\} \tag{9}$$

### 3.3.4 Reconstruction of feature lines

Feature lines of glacier surface were used to express $G_{pl}$, $G_{br}$, $A_l$, $P_{max}$, $P_{min}$, and the intersection of $A_l$ and $G_{pl}$. The schematic diagram of merging the glacier surface features is illustrated in Fig.5.

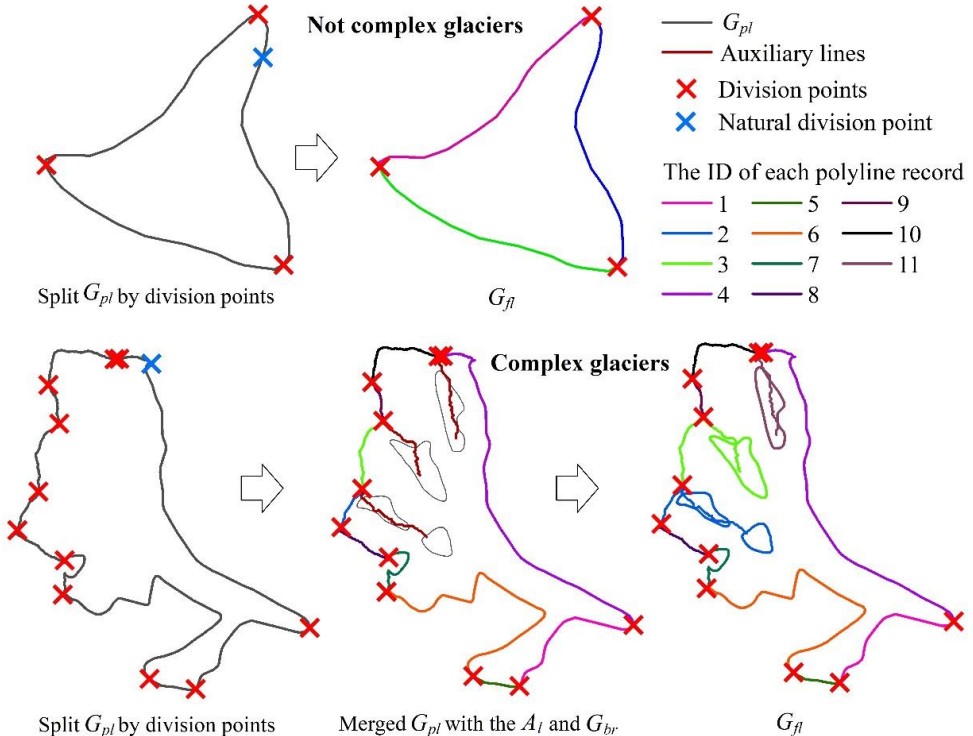

**Figure 5: The schematic of extracting the polyline features of glacier surface.**

For simple glacier and simple compound glacier, it was only necessary to merge $P_{max}$ and $P_{min}$ into a vector file, then split $G_{pl}$, and allocate one unique code for each polyline after converting it from multipart to singlepart. For complex glacier, the processing method was composed of several steps. First of all, the $G_{pl}$ split by division points needed to be combined with $G_{br}$ (if any) and $A_l$ (if any) into a vector file. After converting it from multipart to singlepart, program would allocate again code for each polyline and remark it as $G_{sp1}$. Secondly, polyline records in $G_{sp1}$ were selected one by one with $A_l$, and then the polyline records belonging to the same part in $G_{sp1}$ were merged, which was recorded as $G_{sp2}$. Thirdly, $G_{edge}$ was exported by selecting $G_{sp2}$ using $G_{pl}$, and $G_{alone}$ was exported after switching selection, which represented the bare rock region that still existed independently after merging the glacier outlines with the auxiliary lines. Finally, adopting the proximity algorithm, each element (if any) in $G_{alone}$ was processed in turn with $G_{edge}$. Specifically, it needed three steps: (i) The vertex set $E$ (Eq.10) of $G_{edge}$ and the vertex set $U$ (Eq.11) of $G_{alone}$ were obtained. (ii) The pairs of polylines (Eq.12) matched by serial number were calculated and made the corresponding marks in $G_{sp2}$; (iii) The feature lines ($G_{fl}$) of glacier surface were reconstructed by





merging the same marks in $G_{sp2}$.

$$E_i = \{E_{ij} | j \in [0, card(E)\} \tag{10}$$

$$U_p = \{U_{pq} | q \in [0, card(U)\} \tag{11}$$

$\quad D = \{(p, i) | Min(d(U_p, E_i))\} \tag{12}$

### 3.3.5 Extraction of glacier centerlines

Glacier centerlines ($G_{cline}$) were achieved with the function of Euclidean allocation in arcpy, which needed the input of $G_{fl}$ and

setting the value of $P_8$. The final glacier centerlines ($G_{gcf}$) were obtained by processing $G_{cline}$ with Peak algorithm, after setting

the tolerance for smoothing polylines ($P_{10}$). The schematic diagram of extracting $G_{gcf}$ and the longest length of glaciers ($G_{Lmax}$)

$\quad$ is shown in Fig.6.

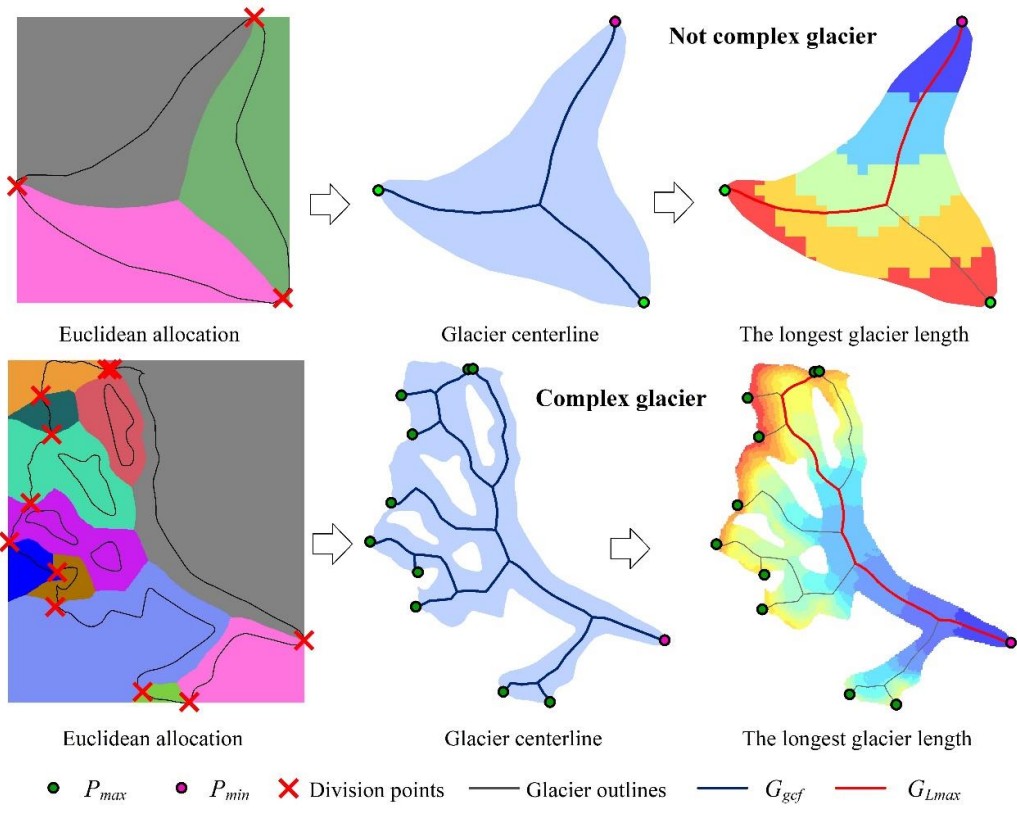

Figure 6: The schematic of extracting glacier centerlines and the longest glacier centerline.



### 3.3.6 Calculation of glacier length

The polyline's code of the end of $G_{gcf}$ was determined by $P_{min}$ after $G_{gcf}$ being converted from multipart to singlepart and

unified coded. Then all branches of glacier centerlines and glacier length were achieved using algorithm (Fig.7) similar to the

critical path. This work consisted of four steps: (i) the polyline set of $G_{gcf}$ was recorded as $C$ (Eq.13), then the sets of polyline

length ($L$) and polyline endpoint ($S$) (Eq.13) were obtained; (ii) the initial search point ($B$) (Eq.14), the end of glacier centerline,

was determined by the coordinates of $P_{min}$ based on the above steps. The common endpoint set ($N$) (Eq.14) with the next parts

of glacier centerlines was obtained, and then the polyline code corresponding to $B$ was recorded; (iii) each element in $N$ was

used as a new starting point for search respectively ($B'$) (Eq.15), which was used to get the common endpoint set ($N'$) (Eq.15)

with the next parts of glacier centerlines. The coding of the corresponding polyline set of each glacier branch was recorded

separately and (vi) the above process continued until all branches of glacier centerline trace back to its corresponding $P_{max}$

(Eq.16).

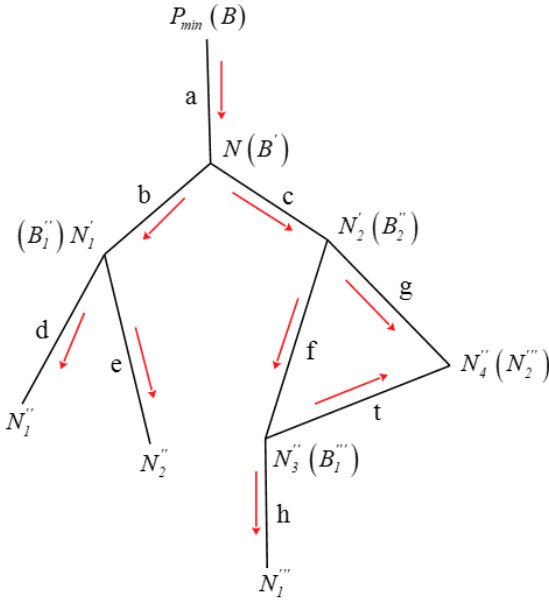

215                                                **Figure 7: The schematic of calculating glacier length.**

$$C_i = \{C_{ij} | j \in [0, card(C)\}$$
$$S = \{(s_i, e_i) | s_i = C_{[i][0]}, and\ e_i = C_{[i][card(C_i)-1]}\}$$

(13)





$$B = \{k | P_{min} \in S_k\}, k \in [0, card(S))$$
$$N = \{P | P \neq P_{min}, and\ P \in S_B\} \tag{14}$$

$$B' = \{k | N \in S_k, and\ k \neq B\}, k \in [0, card(S))$$
$$N' = \left\{P \middle| P \neq N, and\ P \in S_{B'_m}\right\}, m \in [0, card(B'))  \tag{15}$$

$$res = \{\{a,b,d\},\{a,b,e\},\{a,c,f,h\},\{a,c,g\},\{a,c,f,t\}\} \tag{16}$$

The length of each branch of glacier centerlines was counted. The average length (Eq.17) of all branches was called the average

length of a glacier ($G_{Lmean}$). The longest length (Eq.18) of all branches was called the longest length of a glacier ($G_{Lmax}$). In

addition, the part above ELA in $G_{Lmax}$ was regarded as the accumulation region length ($G_{Lacc}$) of a glacier, and the part of $G_{Lmax}$

with altitude lower than ELA was regarded as the ablation region length ($G_{Labl}$) of a glacier. Finally, the corresponding vector

data were generated and some attributes including the corresponding polyline code, glacier code, the value of glacier length

were added.

$$L_{mean} = \frac{SUM(L_{res_i})}{card(res)} \tag{17}$$

$$L_{max} = Max\left(L_{res_i}\right) \tag{18}$$

## 4 Accuracy evaluation and the results

### 4.1 Methods of quality analysis

Here, we used SCGI as the test data to run the designed program, including 48571 glaciers. The extraction results of some

typical examples of glaciers (from simple to complex) are presented in Fig.8. The accuracy of glacier centerlines was evaluated

based on a random verification method in this study. All glaciers (total quantity: $N_G$) corresponding to the samples were

obtained and arranged in ascending order of the area. Specifically, 100 random integers were generated in the set of $[0, N_G]$.

Glaciers with corresponding serial number were exported as samples. After the visual inspection, the accuracy evaluation was

conducted based on the following statistical analysis.





**Figure 8: The centerlines for some typical glaciers.**

Firstly, 100 glaciers were randomly selected from the glacier dataset as samples to obtain a verification accuracy ($R_1$) (Eq.19).

Secondly, each level of glaciers was separately taken as the total ($N_T$), and 100 glaciers were randomly selected. There were 5

samples for 5 levels, which were used to calculate a verification accuracy ($R_2$) (Eq.20) by taking the number proportion of

each glacier level as the weight. Then, 100 glaciers with the largest, middle and smallest areas were selected separately as

samples. The verification accuracy ($R_3$) (Eq.21) was derived using 1:2:1 as the allocation proportion of weight. Finally, the





average value of $R_1$, $R_2$ and $R_3$ was used as the comprehensive accuracy ($R$) (Eq.22). Among them, $S_i$ represented the verification accuracy of the $ith$ sample ($i = \{1,2,3,4,5,6,7,8,9\}$).

$$R_1 = S_1 \tag{19}$$

$$R_2 = \sum_{i=5}^{9} \frac{S_i \times N_{T_i}}{N_G} \tag{20}$$

$$R_3 = 0.25 \times S_2 + 0.5 \times S_3 + 0.25 \times S_4 \tag{21}$$

$$R_3 = \frac{R_1 + R_2 + R_3}{3} \tag{22}$$

**4.2 Sample selection and assessment criteria**

Visual inspection in combination with satellite images and topographic maps is the most direct evaluation method for extraction results. Using 48571 glaciers in China as the test data, nine samples of 900 glaciers were selected for three verifications according to the evaluation method defined in section 4.1. The samples used for verification and relative information are given in Table 2.

**Table 2 The information about validation samples.**

| Verification identifier | 1-whole | | 2-area | | 3-levels | | | | |
|---|---|---|---|---|---|---|---|---|---|
| Sample identifier | a | b | c | d | e | f | g | h | i |
| Selection conditions | Random | Max. | Central | Min. | | | Random | | |
| Sample number | 100 | 100 | 100 | 100 | 100 | 100 | 100 | 100 | 100 |
| Total amount | 48571 | | 48571 | | 38463 | 7341 | 2061 | 501 | 205 |
| Proportion of sample (%) | 0.21 | | 0.62 | | 0.26 | 1.36 | 4.85 | 19.96 | 48.78 |
| Proportion of total (%) | 100 | | 100 | | 79.19 | 15.11 | 4.24 | 1.03 | 0.42 |

Considering the possible defaults of the input data, we set some standards of accuracy evaluation (Table 3). The first level includes three categories: correct (I), inaccurate (II) and incorrect (III). The secondary categories were divided into 11 categories according to probable causes, among which the inaccurate causes and incorrect causes were subclassified as 6 types and 4 types, respectively. Type II involves mostly glaciers with accurate $G_{Lmax}$ but missing, redundant or unreasonable branches of glacier centerlines. When calculating the comprehensive accuracy, category I and II were regarded as correct, and only III was considered incorrect.





**Table 3 The rules of accuracy assessment.**

| 1st-level categories | | 2nd-level categories | |
| --- | --- | --- | --- |
| Code | Descriptions | Code | Descriptions |
| I | Correct | 11 | Correct |
| | | 21 | Inaccurate glacier outlines |
| | | 22 | Inaccurate identification of extreme points |
| | | 23 | Inaccurate proximity algorithm for bare rock regions |
| II | Inaccurate | 24 | The influence of shunt or convergence in the glacier centerlines |
| | | 25 | Inaccurate ridgelines |
| | | 26 | Others (issues that are unknown by the algorithm itself, glaciers or DEM data) |
| | | 31 | Undivided glaciers |
| | | 32 | Ice caps |
| III | Incorrect | 33 | Slope glaciers |
| | | 34 | Others (unknown issues by the algorithm itself, issues with glaciers and DEM data, indistinguishable glacier types, etc.) |

**4.3 Statistics of different samples**

According to the standards in Table 3, the selected samples were conducted with visual investigation. The results of nine

samples were displayed in Fig.9. The statistical results showed that the accuracy of verification-2 was the highest (95.25%),

the accuracy of verification-3 was the second (94.76%) and the accuracy of verification-1 was 93%. The comprehensive

accuracy of glacier centerlines was 94.34%, of which category-I and category-II accounted for 86.06% and 8.28%, respectively.

Meanwhile, we summarized the frequency of each type in each sample basing on 2nd-level categories. As seen in Fig.10, the

problems of centerlines of small glaciers were mainly caused by the inaccurate selection of division points due to the

insufficient accuracy of DEM (code: 22) and incorrect calculation results of some slope glaciers (code: 33). The problems of

centerlines of large glaciers were mainly concentrated in some types coded in 31 and 32, which needed to be repartitioned and

recalculated. In addition, a few problems were found in samples: the upper outlines of glacier were across the ridgeline; a small

number of glaciers were not correctly segmented; the altitude in glaciers' DEM was abnormal. It implied that the reasonable

glacier outlines and accurate DEM data were the prerequisite for extracting glacier centerlines and calculating glacier length.





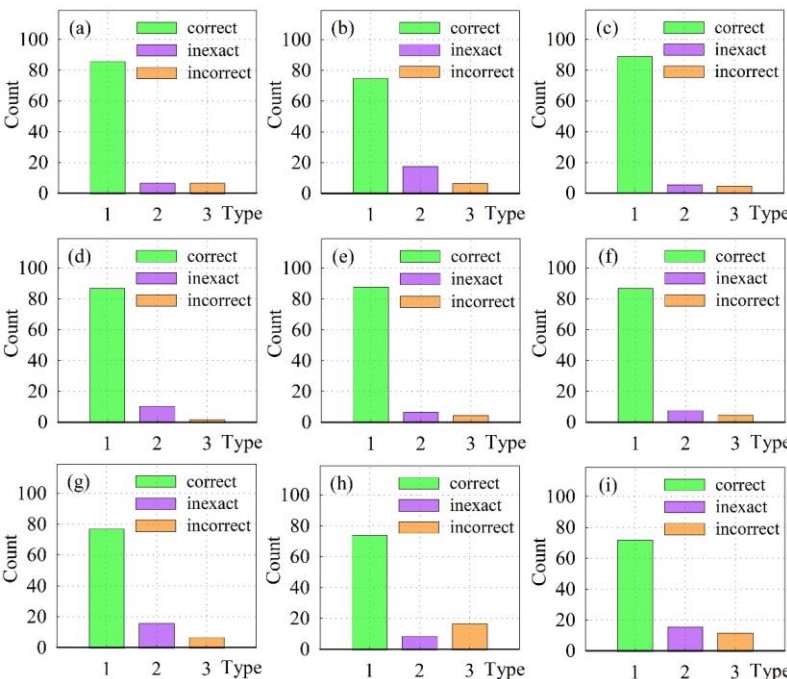

**Figure 9: The statistical chart of evaluating results according to the 1st-level categories.**

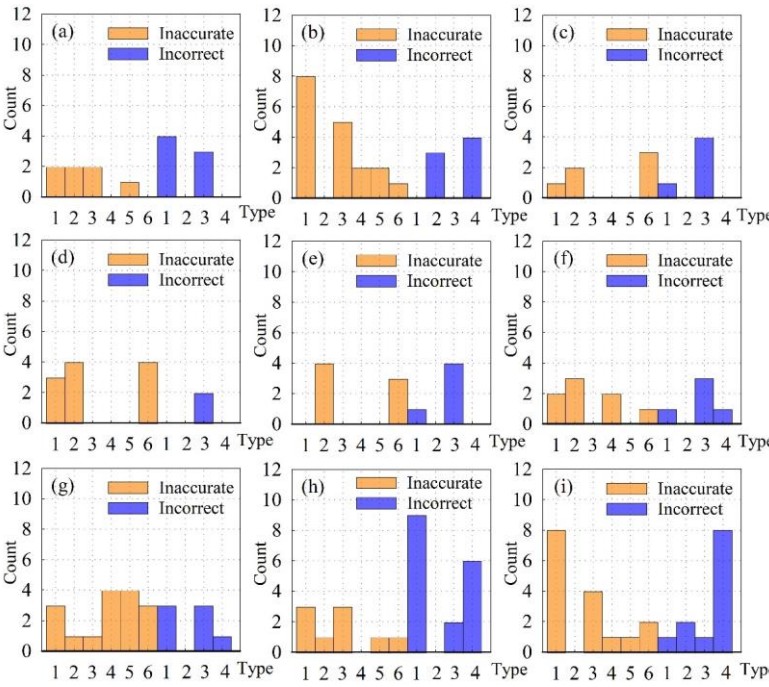

**Figure 10: The statistical chart of evaluating results according to the 2nd-level categories.**



### 4.4 Comparison to glaciers' maximum length from the RGI v6.0

**4.4.1 The statistic of bit order and $D_L$**

In the RGI v6.0, 38053 glaciers in the SCGI were adopted and accounted for 78.35% of the total glaciers in China, by checking

the GLIMS_ID in both glacier datasets. As mentioned above, the $L_{max}$ of the longest glacier length was contained in the RGI

v6.0. In order to further verify the accuracy of glacier length calculated by this method, we calculated the difference ($D_L$)

between $G_{Lmax}$ and $L_{max}$, and then arranged them in ascending order to generate the distribution diagram of sequence-$D_L$ (Fig.11).

If $D_L$ was negative, the $G_{Lmax}$ of glaciers with the corresponding serial number was smaller than $L_{max}$ and vice versa. Overall,

there were only a small part of glaciers with extremely large $|D_L|$ at both ends (Fig.11). After visual inspection, $G_{Lmax}$ was more

consistent with the actual conditions of glaciers.

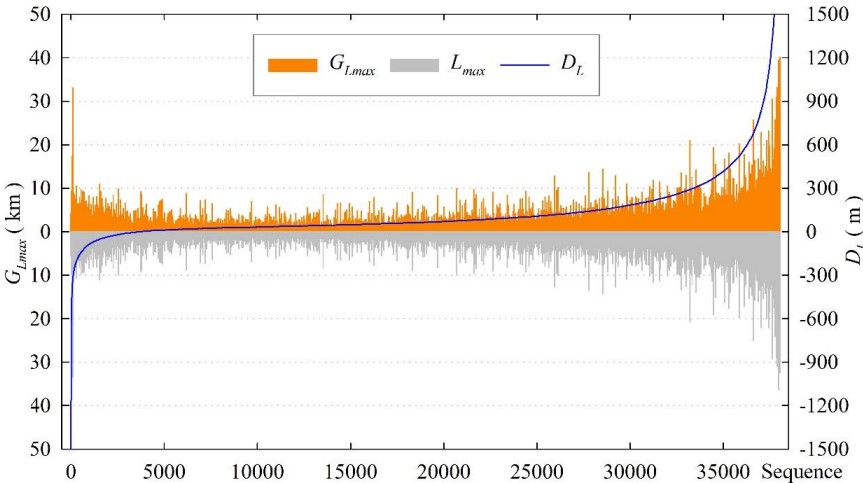

**Figure 11: The statistical chart of DL of the longest glacier length between this dataset and RGI v6.0.**

In addition, the average value of positive $D_L$, the average value of negative $D_L$ and the number of glaciers in different levels

were calculated (Fig.12). The size of three pixels for DEM was used as the statistical tolerance, which means glaciers within

the tolerance range were regarded as consistent extraction results. Statistically, there were 22017 glaciers with equal tolerance,

925 glaciers with negative $D_L$ and 15111 glaciers with positive $D_L$. In terms of numerical comparison, $G_{Lmax}$ obtained by our

method was slightly larger than $L_{max}$ in RGI v6.0.





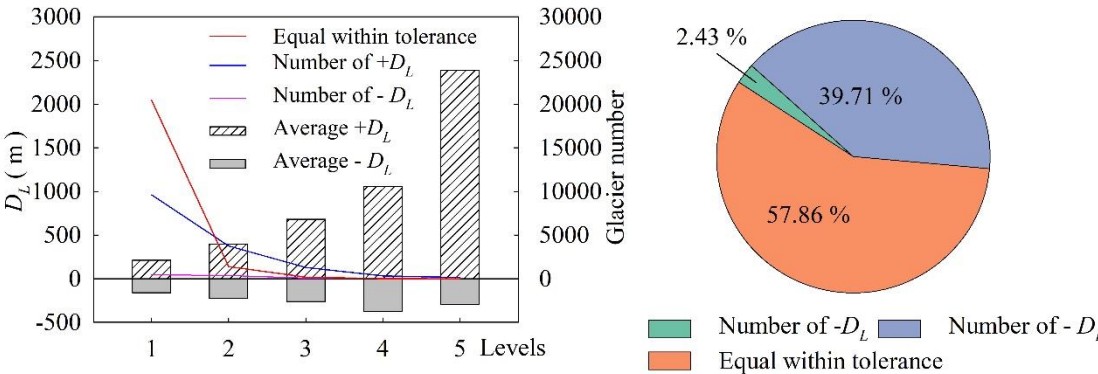

**Figure 12: The statistical charts of DL of the longest length of glaciers by two methods in different glacier scales.**

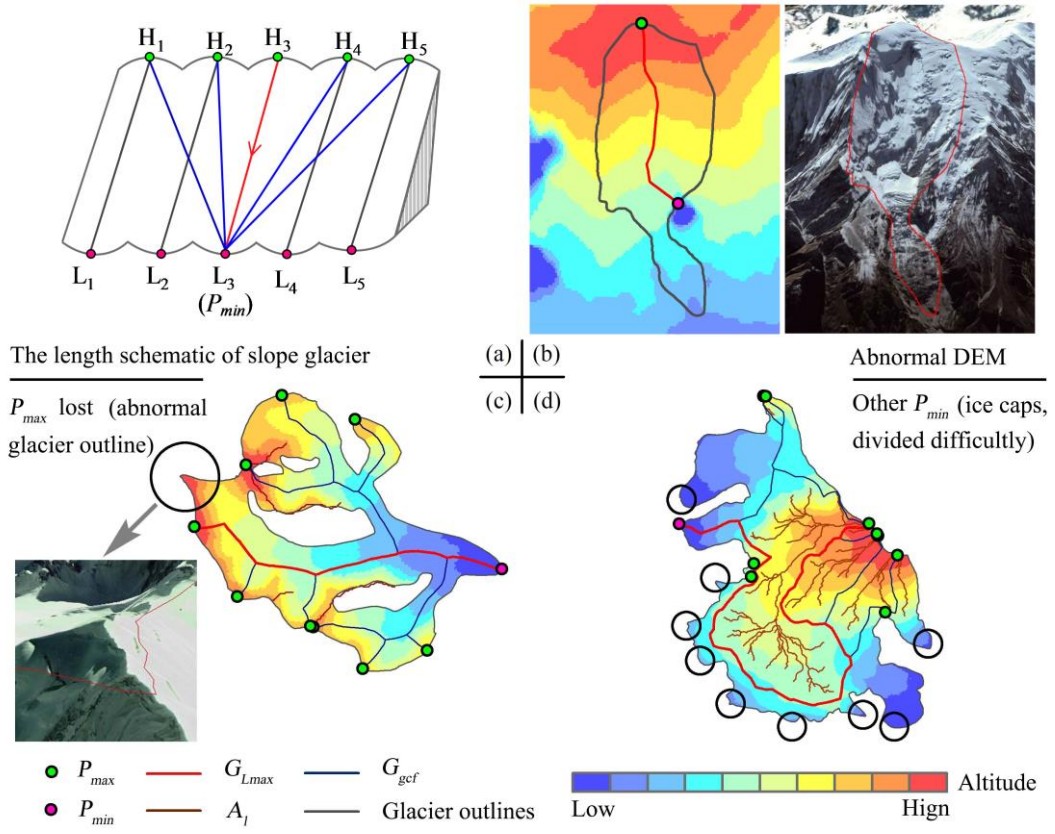

**Figure 13: The schematic of possible causes for the abnormal of the longest glacier length.**



**4.4.2 Analysis of abnormal $D_L$**

Combining the designed algorithm with visual inspection, the preliminary analysis showed that the local abnormal DEM,

inaccurate glacier outlines and some glacier types (such as ice cap, slope glacier, etc.) were the main causes of abnormal $D_L$

(Fig.13). Slope glacier is typical multi-origin and multi-exit glacier with almost the same number of local highest points and

local lowest points, which often exist in pairs (Fig.13-a). If the local highest point did not match the local lowest point, a value

of positive $D_L$ would occur (Fig.13-a, blue polyline). Local abnormalities in DEM generally resulted in a shorter $G_{Lmax}$

(negative $D_L$), as showed in Fig.13-b. Some key local highest points could not be identified because of the inaccurate outlines,

resulting in a large negative $D_L$ (Fig.13-c). For non-single glacier, this algorithm could only identify a lowest point, and all

branches of glacier centerlines converge to this point, which would increase the length of most branches and make $G_{Lmax}$ to be

too large or even wrong (Fig.13-d).

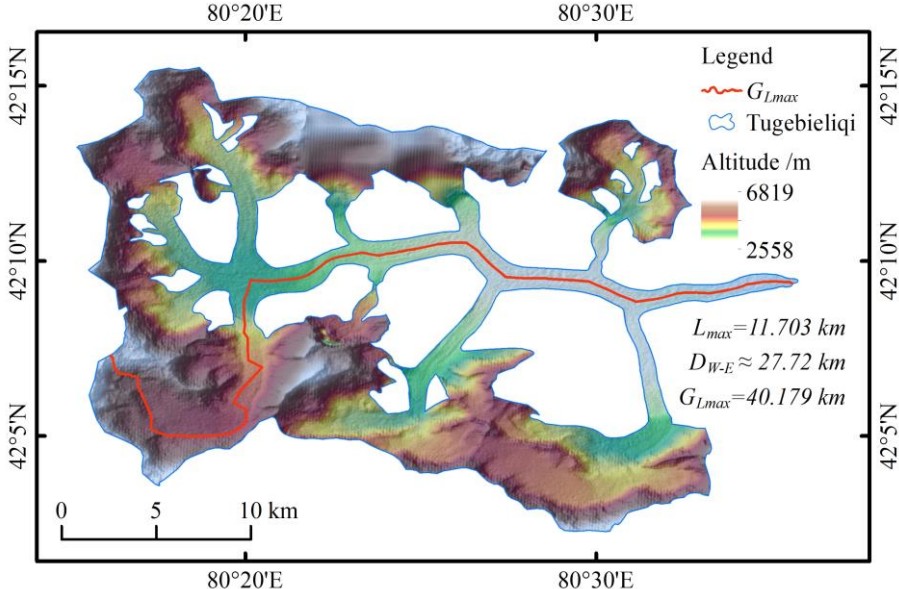

**Figure 14: The schematic of the longest length of Tugebieliqi Glacier.**


The small or abnormal $L_{max}$ of some glaciers was also the main reason of abnormal $D_L$. An abnormal example is shown in

Fig.14. The Tugebieliqi Glacier (GLIMS_ID: G080334E42156N) with the maximum $|D_L|$ is the third largest glacier in China,

only smaller than the Sugatyanatjilga Glacier and the Tuomuer Glacier. Its $G_{Lmax}$ was 40.179 km, but its $L_{max}$ in the RGI v6.0





was only 11.703 km. The further measurement by Google Earth showed that the west-east length ($D_{W-E}$) of the glacier was

about 27.72 km, which meant that our result was more conformable to reality.

## 5 Discussion

### 5.1 Performance of algorithm

In the process of extracting centerlines of glaciers in China, all glaciers were equally divided into eight tasks according to the

number and considering the running efficiency of the algorithm. Based on the actual extraction results, five glaciers that failed

to execute were added as the ninth task. Tasks coded T1~T9 were executed in the working environment of ArcGIS 10.4

software. Except for T7 and T9 using a Lenovo G410 (processors: Intel(R) Core(TM) i5-4210M CPU @ 2.60 GHz; memories:

4GB DDR3L 1600 MHz; video card: AMD Radeon R5 M230 2GB Discrete graphics) of home laptops, the other seven tasks

used seven Dell OptiPlex 7040 (processors:Intel(R) Core(TM) i7-6700 CPU @ 3.40 GHz; memories: 8GB DDR4 2633 MHz;

video card: AMD Radeon(TM) R5 340X 2GB Integrated graphics) of the tower server with the same configuration. The task

distribution and execution results of the tests are given in Table 4.

**Table 4 The statistics of assigning tasks and results of execution in tests.**

| Task ID | Assigned amount | Completed amount | Completion rate (%) | Time (h) | Average time (s) |
|---------|-----------------|------------------|---------------------|----------|------------------|
| T1 | 6000 | 6000 | 100 | 31.00 | 18.60 |
| T2 | 6000 | 6000 | 100 | 29.75 | 17.85 |
| T3 | 6000 | 5999 | 99.98 | 30.53 | 18.32 |
| T4 | 6000 | 6000 | 100 | 29.34 | 17.61 |
| T5 | 6000 | 6000 | 100 | 33.54 | 20.12 |
| T6 | 6000 | 5999 | 99.98 | 31.62 | 18.97 |
| T7 | 6000 | 5999 | 99.98 | 58.63 | 35.18 |
| T8 | 6571 | 6569 | 99.97 | 38.27 | 20.97 |
| T9 | 5 | 5 | 100 | 0.12 | 86.26 |
| Total | 48571 | 48571 | 100 | 282.81 | 20.96 |

The results of the tests showed that the program took an average of 20.96 s to extract a glacier, and it spent 86.26 s or even

longer for some complex glaciers. Among the first eight processing tasks, T4 took the least time. The main reason was that the

assigned glaciers in this task were mostly small and complex glaciers were less, except for the higher machine configuration.

T7 took the longest time, and the cause was the lower machine configuration. The results of all tasks were merged to obtain

the centerline dataset of the SCGI. It contained seven vector files (56 items) and nine logs, which took up about 912 MB in

the storage.

## 5.2 Influence of glacier outline quality and DEM

The extraction method of glacier centerlines belongs to geometric graphic algorithm and depends on glacier outlines. Natively,

comparing with the previous studies, our method has similar problems: (i) the delayed shunt and early convergence of the

branches and (ii) the centerlines of same glacier in different periods, which is not geometrically comparable for some glaciers

in drastic changes of outlines. The extraction results also showed that the branches of some glacier centerlines did have delayed

diversion or early convergence, while the impact on the simulation of glacier's main flowline was limited. Considering that

the results of extracting glacier centerlines change with the changes of glacier outlines, the measurement of the length change

of glaciers in different periods will be the focus of our future work. We may further design a new algorithm to automatically

supplement, extend, delete or modify the benchmarking glacier centerlines, so as to measure the changes of centerlines and

length of glaciers in different periods.

Bare rock region refers to the non-glacial component that is within the outer boundary of the glacier outlines but is not covered

by snow or ice. It can be divided into two types: one is the exposed rock protruding on the glacier surface; the other is the cliff

generally existing between the upper part of the glacier and the firn basin. The snow or ice on the upper part of the glacier

enters the firn basin through the cliffs. And the snow or ice on the cliffs are also important sources of replenishment for firn

basin. So the cliffs are theoretically considered to be part of the glacier. However, the cliffs may be similar to the bare rock

area during the ablation season, and the cliffs are often accompanied by the presence of image shadows, which will easily

cause misjudgments of glacier outlines in interpretation.

Determining the ownership of bare rock regions in $G_{fl}$ will improve the quality of glacier centerlines. In this study, all bare
rock regions were considered to be the first category, and such cases were handled accordingly. The first category was divided

into two types: (i) the bare rock area on the upper part of the glacier being equivalent to the ice divide and (ii) the bare rock

area near the end of the glacier. The attribution of most bare rock areas in the upper part of the glacier can be determined by

the intersection point of $A_l$, $G_{pl}$ with $G_{br}$. Only a few bare rock areas still exist alone, Eq. (12) was required to determine the

segments of the $G_{fl}$ to which they belong. Some bare rock areas located in the ablation area were allowed to exist alone in the

$G_{fl}$, and the probability of their existence was extremely low.

The determination of glacier's ELA is difficult. Some scholars believed that each glacier has its own ELA (Sagredo et al.,

2016;Cui and Wang, 2013), but other scholars argued that the ELA of all glaciers in a certain region is the same (Sagredo et

al., 2014;Jiang et al., 2018). The measurement of ELA requires continuous and long-term observation data, so it is very difficult

to determine the ELA of the glaciers in large-scale. In this study, the ELA used to distinguish between the accumulation area

and the ablation area of the glacier was estimated by calculating the $Z_{min}$. For some glaciers (such as calving glaciers), the $Z_{min}$

is above the actual ELA, which has been reasonably explained by scholars (Braithwaite and Raper, 2009). And it was

considered that this overestimation is unlikely to affect the automatic calculation of glacier length (Machguth and Huss, 2014).

**5.3 Some other factors influencing centerline of glaciers**

Automatic extraction of glacier centerlines was basically carried out during the processing of polylines, so the processing

algorithm of polylines in the program occupied a considerable part of codes. Among them, several common problems of

disconnected polylines are shown in Fig.15. The following four types are important, which have a great influence on the

accuracy and extraction automation of glacier centerlines.

(i) During the post-processing of the auxiliary lines, due to the inaccuracy of ice divide or the problems of DEM, the ridgelines

in the edge of the ice divide of some glaciers start at the $G_{pl}$ and end up with the $G_{pl}$ or in parallel along the $G_{pl}$, which are

unreasonable. In response to this problem, the algorithm set corresponding rules for screening in the processing of auxiliary

lines, reducing the impact of such problems as much as possible.





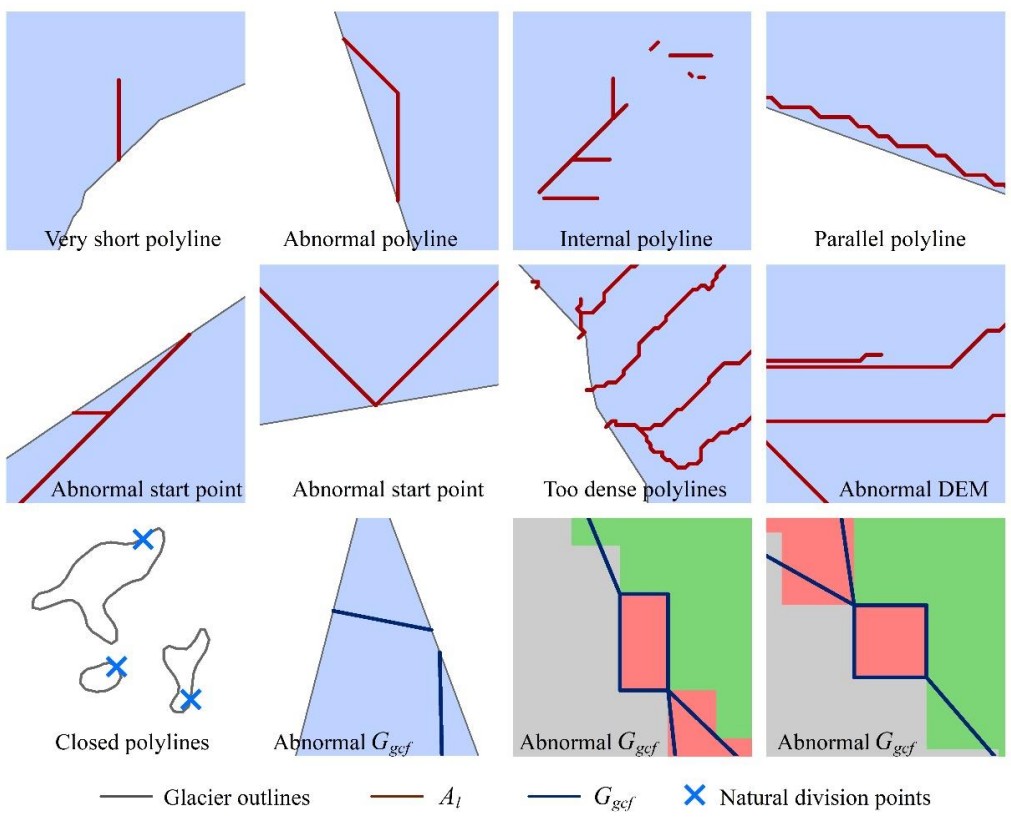

**Figure 15: The schematic of discontinuous short polylines.**

(ii) The visually closed vector polyline is not completely closed. Its start and end are at the same point, which is equal to a natural division point. Unless the natural division point of $G_{pl}$ completely coincides with a certain division point, the number of polyline records in the $G_{pl}$ after division will be one more than we expected. Therefore, it is necessary to identify the natural division point during processing and merge the two disconnected polyline records.

(iii) The algorithm of Euclidean allocation is accomplished based on raster operation, which is equivalent to the equidistant scatter operation with the interval of $P_8$ on the glacier surface. For some glaciers with horizontal or vertical distribution of the $G_{pl}$, the extraction will continue after the centerlines overlaps with the $G_{pl}$. We only need to design the corresponding functions to detect and delete this redundancy of the disconnected polylines.

(iv) In the process of calling the module of Euclidean allocation to generate the centerlines, there is a slight probability that

pixels with strictly equal distances will appear. The central axis will generate a regular rectangle based on the raster pixel

corresponding to the central point, which will affect the calculation of the $G_{Lmax}$. In the algorithm, a function to identify and

deal with such problems was added after the Euclidean allocation, then the polylines on one side of the diagonal of a rectangle

were randomly retained.

**6 Conclusions**

An automatic method for extracting glacier centerlines based on Euclidean allocation in two-dimensional space was designed

and implemented in this study. The automatic extraction method only needs the glacier outlines and the corresponding DEM

to automatically generate the graphic data of $G_{gcf}$, the graphic and numerical data of different glacier lengths ($G_{Lmax}$, $G_{Lmean}$,

$G_{Lacc}$, and $G_{Labl}$). The standardized and automatic extraction of glacier centerlines requires no manual intervention. Meanwhile,

we used SCGI as the test data to run the program. The success rate of extracting glacier centerlines was very close to 100%

and the comprehensive extraction accuracy reached 94.34%, which reflected the robustness and simplicity of our method.

The automatic extraction algorithm proposed has three advantages: (i) introducing the auxiliary reference lines which ensure

the validity of the upper glacier centerlines; (ii) success in automatically obtain the longest length of each glacier and the

branches of glacier centerlines; (iii) providing more information of glacier lengths (e.g., $G_{Lmax}$, $G_{Lmean}$, $G_{Lacc}$, and $G_{Labl}$).

Compared with the $L_{max}$ in RGI v6.0, the $G_{Lmax}$ value calculated by our algorithm is in better agreement with the actual length

of the glacier. Our future work will focus on improving the time efficiency of the algorithm, providing the updated datasets of

glacier centerlines with higher-quality, and identifying the abnormal glacier phenology such as glacier surging rapidly.

**Code availability**

The code used to support the findings of this study are available from the corresponding author upon request.



## Data availability

The datasets including the SCGI, RGI v6.0 and SRTM1 DEM v3.0 used in this study are freely available. The database of glacier centerlines in the SCGI produced in this study are available from the corresponding author upon request.

## Author contribution

Xiaojun Yao designed this algorithm of extracting glacier centerlines and edited the manuscript. Dahong Zhang implemented the program and wrote the first draft of the manuscript. Hongyu Duan tested the program and checked the quality of glacier centerlines. Shiyin Liu, Wanqin Guo, Meiping Sun and Dazhi Li reviewed and edited the manuscript.

## Competing interests

The authors declare that they have no conflict of interest.

## Acknowledgements

This research was funded by the National Natural Science Foundation of China (No.41861013, No.42071089, No.41801052), the Open Research Fund of National Earth Observation Data Center (No.NODAOP2020007) and the Open Research Fund of National Cryosphere Desert Data Center (No. 20D02).

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
