# Peer review of "A new automatic approach for extracting glacier centerlines based on Euclidean allocation"

_The Cryosphere, 2020_

## Short Comment (SC1) · 2 Nov 2020

Dear authors,

thank you for what looks like an interesting new way to compute glacier centerlines.

I would like to bring to your attention that there is an open-source python implementation of the Kienholz et al. 2014 algorithm. The method is available as part of the OGGM modelling framework (https://oggm.org/, Maussion et al., 2019). The implementation and parameters are not "one to one" the same as in Kienholz et al. 2014 but similar enough to allow a fair comparison of your method with previously available ones. I think that your manuscript would gain by making this comparison.

Furthermore, I would like to mention that "Code and results are available upon re-

quest" statements are against this journal's data policies (https://www.the-cryosphere. net/policies/data_policy.html). These practices are detrimental to the review process and to science in general. I would highly recommend to publish your code and center-lines in an openly accessible repository and with a clear software license, in particular if you suggest to use your code instead of previous methods for the generation of the RGI's centerlines.

Best regards,

Fabien Maussion

**References**

Kienholz, C., Rich, J. L., Arendt, A. A., and Hock,R.: A new method for deriving glacier centerlines applied to glaciers in Alaska and northwest Canada, The Cryosphere, 8, 503-519, doi: 10.5194/tc-8-503-2014, 2014.

Maussion, F., Butenko, A., Champollion, N., Dusch, M., Eis, J., Fourteau, K., Gregor, P., Jarosch, A. H., Landmann, J., Oesterle, F., Recinos, B., Rothenpieler, T., Vlug, A., Wild, C. T., and Marzeion, B.: The Open Global Glacier Model (OGGM) v1.1, Geosci. Model Dev., 12, 909–931, https://doi.org/10.5194/gmd-12-909-2019, 2019.

---

## Referee Comment (RC1) · Anonymous Referee #1 · 30 Dec 2020

This is a valuable paper to attempt to get more accurate centerlines in glaciers. The major motivation is to automatically generate high-quality glacier centerlines in a wide range. Authors present and implement a novel automatic algorithm deriving glacier centerlines basing on the feature lines of glacier surface rather than the unit of raster pixel. This method is tested in the second Chinese glacier inventory, and the assessment criteria are built. And the results have been excellent. China has a wide variety of glaciers, so this method should be universal and has a potential application for glacier change.

Overall Quality

Overall, the part of the explanation of the algorithm (Section 3) describes the method in detail, and the flow chart is easy to understand. The research results are convincing,

and some deficiencies, worries and expectations are also reflected in the discussion section. I appreciate that the authors were transparent about the limitations of the method, but have still published what is an interesting study. This study is of great importance to further improve the quality of global glacier centerlines. I recommend it to be published with minor changes.

Specific Comments

Title: I think the title should point out that the author's approach is different from other studies, for example "base on . . ..."

*(P1L20) "the largest length" -> "the longest length" or "the maximum length".

*(P2L30) "Alternatively" might be "Therefore".

*(P2L31) Please add a sentence to explain the role of the two concepts of glacier axis and glacier centerline and their relationship with glacier flowline.

*(P2L45) Delete "automatic". It is too early to mention the importance of automatic extraction algorithm because it cannot be illustrated above.

*(P2L46- P3L60) This section seems not make clear the challenge of current glacier centerlines extraction.

*(P4L80) The provincial boundary is not obvious to see in Figure 1, and the number of map's scale is best such as 100, 200, 500, 1000 km.

*(P5L85) "arcpy" -> "ArcPy"

*(P5L95) Make some parameters clear, for example, PG, A, P, AG.

*(P5L101) Author should explain where the formula 1-3 comes from?

*(P7L124) Some word's fonts in Figure 2 are not uniform. Please check. In addition, I have a question, did DEM need preprocessing? Such as filling.

*(P8L134) median elevation $Z_{min}$ -> median elevation $Z_{med}$. Please check the full

text.

*(P9L144) "the material flow" -> "the mass flow"

*(P9L147) As for post-processing, please introduce in more detail.

*(P12L198) How exactly did the authors get the final glacier centerlines?

*(P14L234) How exactly did the authors visual inspection? Some glacier centerlines may be visually indistinguishable.

*(P19L280) Is the DEM used for maximum length calculation in RGI6.0 same with the author's?

(P24L364) Maybe I missed some details. How did the authors get ELA through Zmin? Maybe the author meant Zmed?

(P26L409) When the article was accepted, I requested the authors to consider making the source code or tool available on Github or some elsewhere.
* * *

---

## Referee Comment (RC2) · Anonymous Referee #2 · 5 Jan 2021

**General Comments:**

This paper automates glacier centerline extraction based on glacier surface features and Euclidean allocation. The author tested their method on the second Chinese glacier inventory and achieve high accuracy and efficiency. The author made a comprehensive estimation of the method's performance. However, some explanation and technical details of the method are needed, which are list below. Given this, I recommend this paper for publication after major revisions with attention to comments.

**Specific Comments:**

1. The author uses numerous abbreviations. It would be easier for readers to follow if the author could apply a list of these abbreviations.

2. It would be better if the author could provide more detailed, necessary explanations in the figure captions. Not all the figures are self-explainable. For instance, in figure 4, why the background elevation maps look differently in the first and second columns? For the DEM in the third column, some areas are masked out. It would be better if the author could explain why and how these areas are masked out.

3. Page 6, Line 118: it would be better if the author could explain more about each rule. For instance, (1) why the local highest points must be higher than ELA? (2) Why a glacier has only one exit? The author also mentioned that this single exit could cause problems (See Figure 13d).

4. Figure 2: (1) Could the author explain about extracting DEM and buffering DEM?

5. Figure 3: what is the difference between  $G_{po}$  and  $G_{pl}$ .

6. Page 8, Line 142: it would be better if the author could provide more information about hydrologic analysis.

7. Page 9, Line 148: About identifying abnormal lines, was it done automatically or manually?

8. Page 9, Line 154: Could the author provide more information about the ergodic algorithms?

9. Page 9, Line 158: Could the author illustrate more about how exactly they screen auxiliary lines using P4, P5, and P11?

10. Figure 5: the definition of Natural division point is missing. It would be better if the author could provide an example showing the natural division point.

11. Page 12, Line 196: It would be better if the author could explain more about how Euclidean allocation could get glacier centerlines from  $G_{fl}$ . To me, the Euclidean is the key part for extracting the glacier centerlines. So I think it is worthwhile to illustrate more about it.

12. Page12, Line 197: The author uses the Peak algorithm to get the final glacier centerlines from the glacier centerline. What is the purpose of this step? Figure 2 shows that the Peak algorithm is to smooth the polyline. Why do we need to smooth the polyline?

13. Page 19, Line 287: It would be better if the author could provide examples showing that their results are more consistent with the actual conditions of glaciers comparing with RGI v6.0.

14. Page 19, Line 291: The tolerance here is 90 meters (3 pixels of DEM). It would be better if the author could explain why they choose this value.

15. Page 19, Line 292: If I understand it correctly, I suggest the author rephrase the sentence as "There were 22017 glaciers within the tolerance, 925 glaciers with negative DL and 15111 glaciers with positive DL that are out of the tolerance."

16. Figure 13: For 13b, c, and d, where is the correct glacier centerline? Also, in figure 13d, what do these black circles mean? Please add more information in the figure caption (See comment 2).

17. Figure 15: Please add legends of regions with different colors or illustrate them in the figure caption. Please consider numbering each subfigure (See comment 2).

18. Figure 15: In the first two figures of the second row, where is the abnormal start point? In the fourth figure of the second row, why it is due to abnormal DEM?

**Technical corrections:**

Page 5, Line 85: Consider to change "arcpy" to "ArcPy"

Figure 2: In the part of the extraction of centerlines, it seems that  $G_{cline}$  and  $G_{gcf}$  should switch their position according to the author's definition.

Page 9, Line 148: In the third part of the post-processing, is it "numbers" or "members"?

Figure 9: Consider changing "inexact" to "inaccurate" for consistency.

Figure 12: Consider changing "DL" to " $D_L$ " in the figure caption. For the figure on the right-hand side, the blue color should represent the number of  $+D_L$ , if I understand it correctly.

---

## Author Comment (AC1) · 7 Feb 2021

**Responses to Reviewer #1 to manuscript TC-2020-294**

Thanks for your helpful comments to improve this manuscript.

**Please Notes**: Text in BLACK is the reviewer' comments and our responses are in BLUE.

**Specific Comments:**

Title: I think the title should point out that the author's approach is different from other studies, for example "base on …".

**Response:** We changed the title to "A new automatic approach for extracting glacier centerlines based on Euclidean allocation", which can reflect that our approach is different from other studies.

*(P1L20) "the largest length" -> "the longest length" or "the maximum length".

**Response:** We revised it to "the longest length".

*(P2L30) "Alternatively" might be "Therefore".

**Response:** It has been modified.

*(P2L31) Please add a sentence to explain the role of the two concepts of glacier axis and glacier centerline and their relationship with glacier flowline.

**Response:** We have further explained the related concepts involved in the question:

Glacier centerline is a central line close to the main flowline of glacier, which can be acquired base on glacier axis and be used to simulate the glacier flowline.

In addition, explanations of the relationship of some related concepts are shown in Figure A1.

[Figure]

**Figure A1: The schematic of the relationship of some related concepts.**

*(P2L45) Delete "automatic". It is too early to mention the importance of automatic extraction algorithm because it cannot be illustrated above.

**Response:** This word was deleted.

*(P2L46- P3L60) This section seems not make clear the challenge of current glacier centerlines extraction.

**Response:** So far, the biggest challenge for glacier centerline extraction is still automation. In the past, glacier length was determined manually in a laborious way. In recent years, several authors mentioned in the section have tried to extract the centerlines in batches, however, the results are not satisfactory. In this regard, we added the following summary:

So, the current biggest challenge is still the implementation of automation extraction of glacier centerline and the acquirement of more information about glacier length.

*(P4L80) The provincial boundary is not obvious to see in Figure 1, and the number of map's scale is best such as 100, 200, 500, 1000 km.

**Response:** The Figure 1 was remapped.

*(P5L85) "arcpy" -> "ArcPy"

**Response:** It has been modified.

*(P5L95) Make some parameters clear, for example, $P_G$, $A$, $P$, $A_G$.

**Response:** We rewrote acronym of each parameter to clarify their meanings, listed in the Appendix A. The relevant parameters are explained as follows:

**Table A1 The list of main acronyms in this study.**

| Acronyms | Description |
| --- | --- |
| $A_t$ | The given area of an equilateral triangle |
| $A_g$ | The polygon's area of the glacier's outer boundary |
| $A_l$ | The final auxiliary line |
| $A_r$ | The ridgelines of the glacier surface |
| $G_{br}$ | The bare rock in glacier |
| $G_{fcl}$ | The final glacier centerline |
| $G_{fl}$ | The feature lines of glacier surface |
| $G_{cl}$ | The original glacier centerline |
| $G_{Labl}$ | The length in the ablation region of the glacier |
| $G_{Lacc}$ | The length in the accumulation region of the glacier |
| $G_{Lmax}$ | The longest length of the glacier |
| $G_{Lmean}$ | The average length of the glacier |
| $G_{pl}$ | The polyline of the outer boundary of the glacier |
| $G_{po}$ | The polygon of the outer boundary of the glacier |
| $L_{max}$ | The longest glacier length of RGI v6.0 |
| $D_L$ | The difference between $G_{Lmax}$ and $L_{max}$ |
| $P_t$ | The given perimeter of an equilateral triangle |
| $P_g$ | The perimeter of the glacier's outer boundary |
| $P_{max}$ | The local highest point of glacier outline |
| $P_{min}$ | The lowest point of glacier outline |
| $RGI$ | The Randolph Glacier Inventory |
| $SCGI$ | The Second Chinese Glacier Inventory |
| $Z_{med}$ | The median elevation of the glacier |

Please note that in Table A1, the parameters $A_t$, $P_t$, $A_g$ and $P_g$ correspond to $A$, $P$, $A_G$ and $P_G$ in the manuscript, respectively. The four parameters involved in the comment are explained as follows:

$A_t$ $(A)$: The given area of an equilateral triangle;

$P_t$ $(P)$: The given perimeter of an equilateral triangle;

$A_g$ $(A_G)$: The polygon's area of the glacier's outer boundary;

$P_g$ $(P_G)$: The perimeter of the glacier's outer boundary.

*(P5L101) Author should explain where the formula 1-3 comes from?

**Response:** Formula 1-3 are proposed in this study. Formula 1 expresses the relationship between the perimeter and the area of an equilateral triangle. Formula 2 represents the method for determining the glacier grade in this study. Formula 3 expresses the proportional coefficients for determining the relevant parameters of different levels based on the aspect ratio of the equilateral triangle corresponding to the area of glacier's outer outline.

The main basis is the classification of glacier scale and the scale of glaciers is divided into 12 levels in the SCGI. The values of classification intervals are 0.1, 0.5, 1, 2, 5, 10, 20, 50, 100, 200 and 300 $km^2$. Combined with the sensitivity of the algorithm to each grade of glaciers during the experiment, this research divides the glaciers into 5 grades (interval value: 1, 5, 20 and 50 $km^2$). In the experiment, we also found that when the outer perimeter of glaciers ($P_g$) of the same scale differs greatly, the extraction results of glacier centerlines differ greatly. In addition, the shape of alpine glacier resembles a triangle. Therefore, the $P_g$ was considered in the glaciers' classification in this study, and the classification results were fine-tuned according to the above three formulas with reference to the values of the SCGI's grading intervals.

*(P7L124) Some word's fonts in Figure 2 are not uniform. Please check. In addition, I have a question, did DEM need preprocessing? Such as filling.

**Response:** It has been checked that Figure 2 includes two fonts. The main body of the flow chart uses the Times New Roman (nine pounds) and the module name uses the Microsoft Elegant Black (10 pounds).

Figure 2 briefly shows the processing for DEM. The actual processing includes a series of preprocessing such as clipping, filling, condition selection, focus statistics, and inverse terrain calculations.

*(P8L134) median elevation $Z_{min}$ -> median elevation $Z_{med}$. Please check the full text.

**Response:** It has been modified.

*(P9L144) "the material flow" -> "the mass flow".

**Response:** It has been modified.

*(P9L147) As for post-processing, please introduce in more detail.

**Response:** Firstly, the ridgelines of the glacier surface ($A_r$) were obtained by clipping the ridge lines using $G_{po}$. The set of all possible starting points of auxiliary lines was gained by intersecting $A_r$ with $G_{pl}$. Then, the ridgeline clusters connected to each starting point were achieved and marked by traversing the point set. The number of auxiliary lines was initially determined. Finally, the longest length of each auxiliary line was calculated by adopting the critical path algorithm. The final auxiliary lines ($A_l$) were obtained by screening all auxiliary lines using the three parameters of $P_4$, $P_5$ and $P_{11}$.

The related processing methods are explained in the P10L153- P11L159 of the manuscript. The

processing objects (the disconnected lines and the abnormal lines) of steps i and ii are shown in the discussion section (Figure 15). The post-processing of steps iii, iv and v are shown in Figure A2 in more detail.

[Figure]

**Figure A2: The schematic of post-processing. (a) Before pre-processing; (b) After pre-processing. A total of nine line-clusters are removed by screening.**

*(P12L198) How exactly did the authors get the final glacier centerlines?

**Response:** Firstly, the feature polylines ($G_{fl}$) after automatically deriving by the program are input, and the function of Euclidean allocation in ArcPy is called to generate the division glacier surface. Then the common edges between regions on the dividing glacier surface are identified. Finally, the common edges are automatically checked and processed (including smoothing process) to obtain the corresponding vector data. This study regards them as the final glacier centerlines.

*(P14L234) How exactly did the authors visual inspection? Some glacier centerlines may be visually indistinguishable.

**Response:** The method of visual inspection is detailed in section 4.2 (Sample selection and assessment criteria). Indeed, we also found this problem, however, it is hard to avoid. This research is based on a 2D algorithm. Theoretically, the extraction result of the glacier centerline is correct as long as it meets its definition. Nevertheless, we still loaded it on Google Earth for inspection. In addition, we compared it with the glacier length in the RGI v6.0, and further evaluated the extraction results of glacier centerlines.

*(P19L280) Is the DEM used for maximum length calculation in RGI6.0 same with the author's?

**Response:** We all used SRTM DEM to calculate the longest length of the glaciers. The difference

is the spatial resolution of SRTM DEM (this study: 30 m; RGI v6.0: 90 m).

*(P24L364) Maybe I missed some details. How did the authors get ELA through $Z_{min}$? Maybe the author meant $Z_{med}$?

**Response:** We thank the reviewer for the comment. ELA is estimated by $Z_{med}$, and the relevant content has been corrected above.

*(P26L409) When the article was accepted, I requested the authors to consider making the source code or tool available on Github or some elsewhere.

**Response:** We agree to you. We will provide an executable file and test results if the paper can be published.

---

## Author Comment (AC2) · 7 Feb 2021

**Responses to Reviewer #2 to manuscript TC-2020-294**

Thank you very much for your helpful comments to improve this manuscript.

**Please Notes:** Text in BLACK is the reviewer' comments and our responses are in BLUE.

**Specific Comments:**

1. The author uses numerous abbreviations. It would be easier for readers to follow if the author could apply a list of these abbreviations.

**Response:** Thanks for your suggestion. We added a list (Appendix A: Table A1) of main acronyms at the end of this paper.

**Table A1 The list of main acronyms in this study.**

| Acronyms | Description |
|---|---|
| $A_t$ | The given area of an equilateral triangle |
| $A_g$ | The polygon's area of the glacier's outer boundary |
| $A_l$ | The final auxiliary line |
| $A_r$ | The ridgelines of the glacier surface |
| $G_{br}$ | The bare rock in glacier |
| $G_{fcl}$ | The final glacier centerline |
| $G_{fl}$ | The feature lines of glacier surface |
| $G_{cl}$ | The original glacier centerline |
| $G_{Labl}$ | The length in the ablation region of the glacier |
| $G_{Lacc}$ | The length in the accumulation region of the glacier |
| $G_{Lmax}$ | The longest length of the glacier |
| $G_{Lmean}$ | The average length of the glacier |
| $G_{pl}$ | The polyline of the outer boundary of the glacier |
| $G_{po}$ | The polygon of the outer boundary of the glacier |
| $L_{max}$ | The longest glacier length of RGI v6.0 |
| $D_L$ | The difference between $G_{Lmax}$ and $L_{max}$ |
| $P_t$ | The given perimeter of an equilateral triangle |
| $P_g$ | The perimeter of the glacier's outer boundary |
| $P_{max}$ | The local highest point of glacier outline |
| $P_{min}$ | The lowest point of glacier outline |
| $RGI$ | The Randolph Glacier Inventory |
| $SCGI$ | The Second Chinese Glacier Inventory |
| $Z_{med}$ | The median elevation of the glacier |

2. It would be better if the author could provide more detailed, necessary explanations in the figure captions. Not all the figures are self-explainable. For instance, in figure 4, why the background elevation maps look differently in the first and second columns? For the DEM in the third column,

some areas are masked out. It would be better if the author could explain why and how these areas are masked out.

**Response:** Thanks a lot for your comments. We renamed some figures in the manuscript.

[Figure]

**Figure 3: The schematic of processing raw data ($G_{po}$ denotes the polygon of the glacier; $G_{pl}$ denotes the polyline of glacier's outer boundary; and $G_{br}$ denotes the boundary of the bare rock in glacier).**

[Figure]

**Figure 4: The schematic of extracting auxiliary lines. (a) and (d) demonstrate the digital elevation model (DEM) around the glacier; (b) and (e) show the ridgelines in region covered by DEM; (c) and (f) show the auxiliary lines in glacier.**

[Figure]

(a) Euclidean allocation     (b) Glacier centerlines     (c) The longest glacier centerlines

**Not complex glacier**

(d) Euclidean allocation     (e) Glacier centerlines     (f) The longest glacier centerlines

**Complex glacier**

● $P_{max}$    ● $P_{min}$    ✕ Division points    —— Glacier outlines    —— $G_{fcl}$    —— $G_{Lmax}$

**Figure 6: The schematic of extracting centerlines and the longest centerline of the glacier. (a) and (d) show the results after executing the European allocation, and the different colors represent the regions which have the shortest distance to the corresponding edges of the glacier; (b) and (e) represent the centerlines($G_{fcl}$), the local highest point ($P_{max}$) and lowest point ($P_{min}$) of the glacier; (c) and (f) demonstrate the longest centerline ($G_{Lmax}$) of the glacier and the background is the digital elevation model with the graduated red (high)– blue (low) color.**

[Figure]

**Figure 7: The schematic of calculating glacier length (The red arrow represents the search direction of the branches of glacier centerline).**

[Figure]

**Figure 8: The centerlines for some typical glaciers ($P_{max}$ and $P_{min}$ denote the local highest point and lowest point in the boundary of the glacier, respectively; $A_l$ denotes the auxiliary lines; $G_{fcl}$ and $G_{Lmax}$ denote the centerlines and the longest centerline of the glacier).**

[Figure]

**Figure 13: The schematic of probable causes for the abnormal of the longest glacier length. In Figure b, the red dashed line indicates the revised glacier centerline, and the yellow point is the correct lowest point ($P_{min}$). In Figure c, the red dashed line represents the missing branch, and the yellow point is a local highest point ($P_{max}$) missed by the algorithm. In Figure d, the black circle indicates some probable exits of the glacier, which needs to be divided into individual glaciers before extracting the centerlines.**

[Figure]

**Figure 14: The schematic of the longest centerline of the Tugebieliqi Glacier ($L_{max}$: the corresponding length of this glacier in the RGI v6.0; $D_{W-E}$: the distance from west to east of this glacier; $G_{Lmax}$: the length calculated by our method).**

[Figure]

**Figure 15: The schematic of discontinuous short polylines. Subgraphs a-h represent type (i), i represents type (ii), j represents type (iii) and k-l represent type (iv). The background in subgraphs a-h and j represent glacier-covered areas. Subgraph i shows several closed polylines, which does not fill background color. The different background colors in subgraphs k-l represent different areas of the glacier surface after the European allocation.**

3. Page 6, Line 118: it would be better if the author could explain more about each rule. For instance, (1) why the local highest points must be higher than ELA? (2) Why a glacier has only one exit? The author also mentioned that this single exit could cause problems (See Figure 13d).

**Response:** This paper takes the four rules as the preconditions for the implementation of the algorithm, and clarifies that the processing unit of the algorithm is an individual glacier polygon instead of a no divided glacier such as the ice sheet ice cap. The detailed explanation is as follows:
(i) As glacier heads, the local highest points are typically located at higher elevations. It is generally considered to be higher than the altitude of 1/3 (Kienholz et al., 2014), or 1/2 glacier area, and Median area altitude (latter: $Z_{med}$) can be approximated as ELA (Machguth and Huss, 2014).
(ii) This study assumes that all glacier polygons are correctly divided into single glaciers, that is, there is only one glacier terminus (exit). It is generally considered to be the lowest point of the polyline of the outer boundary of a glacier.

(iii) The auxiliary polyline is used to intervene in the generation of centerline for the upper part of a glacier, so it only acts on the accumulation region of glaciers.

(iv) The feature polylines of the glacier surface are composed of the polylines of the outer boundary of the glacier, auxiliary polylines, and the boundary of the bare rock area, which together determine the flow direction of a glacier centerline.

In addition, the reason for this problem (Figure 13d) is that the glacier was not divided into a single glacier in the Second Chinese Glacier Inventory.

4. Figure 2: (1) Could the author explain about extracting DEM and buffering DEM?

**Response:** In the flow chart of this research, boxes represent the process and the parallelograms represent the results. Therefore, as show as in the flow chart, "buffering DEM" is obtained in the process of "clipping DEM". Specifically, "extracting DEM" refers to the clipping of the DEM, which appeared twice: one is to use the buffering polygon of the outer boundary of the glacier to clip DEM to obtain the "buffering DEM". Its purpose is to extract the feature information of the glacier such as the lowest point, the local highest points and the auxiliary lines; the other is to use the glacier polygon to extract DEM to estimate the ELA.

5. Figure 3: what is the difference between $G_{po}$ and $G_{pl}$.

**Response:** We added the more complete captions in Figure 3:

Figure 3: The schematic of processing raw data ($G_{po}$ denotes the polygon of the glacier; $G_{pl}$ denotes the polyline of glacier's outer boundary; and $G_{br}$ denotes the boundary of the bare rock in glacier).

$G_{po}$ denotes one glacier in 2D geometry (i.e., polygon), and $G_{pl}$ denotes one glacier in 1D geometry (i.e., polyline). Specifically, $G_{po}$ represents the polygon of the outer boundary of the glacier, and $G_{pl}$ refers to the polyline of the outer boundary of the glacier. To identify them more clearly, we collected them in the list (Appendix A: Table A1), as shown as the response to Comment 1.

6. Page 8, Line 142: it would be better if the author could provide more information about hydrologic analysis.

**Response:** For a more detailed presentation, we added the workflow of hydrological analysis (the shaded region in Figure A2) and changed the relevant description in the manuscript as follows:

Based on the inverse terrain method, the extraction of ridgelines was easily accomplished by the workflow of hydrologic analysis.

[Figure]

**Figure A2: The flow chart of the auxiliary line. The workflow of hydrological analysis is shown in the shaded region.**

7. Page 9, Line 148: About identifying abnormal lines, was it done automatically or manually?

**Response:** It is identified automatically by the program. The whole process of the glacier centerlines extraction is no one intervened from data input to results generation.

8. Page 9, Line 154: Could the author provide more information about the ergodic algorithms?

**Response:** This part is a detailed explanation of the five steps of post-processing the ridgelines. The ergodic algorithms are shown in Figure A3, which specifically reflects in the following two aspects: (i) Given a starting point from the set of all possible starting points of auxiliary lines, all the ridgelines of corresponding glaciers are traversed to determine the line cluster composed of the polylines directly or indirectly connected to it.

(ii) Given a line cluster, all the polylines that make up the line cluster are traversed to find the longest ridgeline starting from the starting point.

[Figure]

**Figure A3: The diagram of the application of the traversal algorithms in the part. (a) 14 line clusters in the figure are identified from all polylines; (b) Line cluster eight consists of five polylines, the longest one is [a, d, e].**

9. Page 9, Line 158: Could the author illustrate more about how exactly they screen auxiliary lines using $P_4$, $P_5$, and $P_{11}$?

**Response:** $P_4$ is used to control the shortest auxiliary line, filtering some extremely short auxiliary lines. Only the longest length of the line cluster is less than $P_4$ can be retained.

$P_5$ is used to filter some extremely long auxiliary lines, and the given threshold is relatively large. It has a particularity and is mainly aimed at some narrow and long glaciers.

$P_{11}$ acts as a switch. When the perimeter of the glacier's outer boundary is greater than the value, parameter $P_5$ will be used.

10. Figure 5: the definition of Natural division point is missing. It would be better if the author could provide an example showing the natural division point.

**Response:** In order to make readers better understand the natural division point in this study, we added a schematic of the natural division point, which is shown in Figure A4. It is determined by the storage structure of the closed polyline. It is assumed that there is a set of coordinates [a, b, c, d, e, f, g], which represents the vertex set ($V$) of a polyline ($L$). If $L$ is a closed polyline, then g's coordinate of the last member of $V$ is equal to a's coordinate. Although these two coordinates represent the same position and $L$ is also closed visually, a (the polyline head) and g (the polyline end) are separated in data storage. A breakpoint is formed between a and g, which is the natural division point in this paper.

[Figure]

[a, b, c, d, e, f, g]

*V*          *Structure of L*          *The natural division point of L*

**Figure A4: The schematic of the natural division point.**

11. Page 12, Line 196: It would be better if the author could explain more about how Euclidean allocation could get glacier centerlines from $G_{fl}$. To me, the Euclidean is the key part for extracting the glacier centerlines. So I think it is worthwhile to illustrate more about it.

**Response:** We added more related descriptions, as following:

Original glacier centerlines ($G_{cl}$) were achieved with the function of Euclidean allocation in ArcPy, which needed the input of $G_{fl}$ and setting the value of $P_8$. Firstly, the feature lines ($G_{fl}$) after automatically deriving by the program are input, and the function of Euclidean allocation in ArcPy is called to generate the division glacier surface. Then the common edges between regions on the dividing glacier surface are identified. Finally, the common edges are automatically checked and processed to obtain $G_{cl}$.

The function of Euclidean allocation in ArcPy is used to calculate the nearest source for each cell based on Euclidean distance. It can be divided into three steps:

(i) As the input source locations, $G_{fl}$ is converted to the grid format with a spatial resolution of $P_8$ according to the ID of the polyline clusters;

(ii) The last step also generated a grid data with an extent of the bounding box of $G_{fl}$ and a spatial resolution of $P_8$, which is equivalent to an equidistant scatter array, and can be used as the output source locations;

(iii) By calculating the Euclidean distance between each output source location and the all input source locations one by one, the closest input source (polyline cluster) is determined, and its ID is assigned as the value of the output source location. The raster consisting of the updated output source locations is then exported, that is, the raster of glacier surface after segmented by the function of Euclidean allocation.

12. Page12, Line 197: The author uses the Peak algorithm to get the final glacier centerlines from

the glacier centerline. What is the purpose of this step? Figure 2 shows that the Peak algorithm is to smooth the polyline. Why do we need to smooth the polyline?

**Response:** Glacier centerline ($G_{fcl}$) represents the main flow line of a glacier. The smoothing algorithm can eliminate the zigzag pattern or irregular polylines in the result to make it closer to the actual main flow line of a glacier. This is consistent with the processing methods of the other two related papers (Kienholz et al., 2014, Machguth and Huss, 2014). The Peak algorithm selected in this study is as same as that adopted by Kienholz et al., which is relatively simple and has a better smoothing effect. The difference is that in this study, the zigzag pattern or irregular polylines are caused by the lower spatial resolution of Euclidean allocation (depending on $P_8$: to trade-off the extraction efficiency and the accuracy of the results), while in their study, those are caused by the low-quality DEM.

Meanwhile, the risk of filtering is also very little, because the filtering result (shorter part) is always consistent with the forward trend of glacier centerline.

13. Page 19, Line 287: It would be better if the author could provide examples showing that their results are more consistent with the actual conditions of glaciers comparing with RGI v6.0.

**Response:** There are two reasons why the results of this study are more consistent with the actual conditions of glaciers: (i) For some glaciers with large differences in the longest length of glaciers extracted by the two algorithms, visual inspection can reveal this conclusion. (ii) The spatial resolution of DEM used in this research is better than the RGI v6.0. Correspondingly, the results are more consistent with the actual conditions of glaciers. As an example, figure 14 can reflect this result to some extent.

In the past two years, we have been looking for a set of existing graphical data of glacier centerlines that can be used for verification for this study. We also tried to ask the authors of related papers for help in the form of E-mail. However, we did not get any available information.

Fortunately, RGI v6.0 provides the numerical data of the longest length of glaciers, and most of the corresponding glacier polygons are derived from SCGI. We used the field of GLIMS_ID shared by the two sets of data for matching, and finally obtained the set of numerical data that was used to verify the results of this research.

14. Page 19, Line 291: The tolerance here is 90 meters (3 pixels of DEM). It would be better if the author could explain why they choose this value.

**Response:** The main reason for choosing 3 DEM pixels (90 m) as the tolerance is that the spatial resolution of DEM used to calculate the longest glacier length in RGI v6.0 is 90 meters. Another reason is that in this study, whether the selection of the local highest points or the process of the Euclidean allocation (given maximum P8: 30 m), at least 3 pixels are needed to determine a local highest point or an effective vertex of glacier centerline.

Therefore, this is based on the theoretical maximum error of this study and the minimum error of the longest glacier length in RGI v6.0 as the tolerance of statistics.

15. Page 19, Line 292: If I understand it correctly, I suggest the author rephrase the sentence as "There were 22017 glaciers within the tolerance, 925 glaciers with negative $D_L$ and 15111 glaciers with positive $D_L$ that are out of the tolerance.

**Response:** It has been modified.

16. Figure 13: For 13b, c, and d, where is the correct glacier centerline? Also, in figure 13d, what do these black circles mean? Please add more information in the figure caption (See comment 2).

**Response:** We added more information to the caption of Figure 13:

Figure 13: The schematic of probable causes for the abnormal of the longest glacier length. In Figure b, the red dashed line indicates the revised glacier centerline, and the yellow point is the correct lowest point ($P_{min}$). In Figure c, the red dashed line represents the missing branch, and the yellow point is a local highest point ($P_{max}$) missed by the algorithm. In Figure d, the black circle indicates some probable exits of the glacier, which needs to be divided into individual glaciers before extracting the centerlines.

The revised Figure 13 is shown in Figure A5. We added the correct centerline and the lowest point in subgraph b, and added the local highest points missed by the algorithm and the revised glacier centerline to subgraph c. At the same time, we updated the legend in the new figure.

[Figure]

**Figure A5: The revised Figure 13.**

17. Figure 15: Please add legends of regions with different colors or illustrate them in the figure caption. Please consider numbering each subfigure (See comment 2).

**Response:** The revised Figure 15 is shown in Figure A6. We ranked the 12 sub-graphs from a to l in the revised Figure 15 and added more detailed caption:

Figure 15: The schematic of discontinuous short polylines. Subgraphs a-h represent type (i), i represents type (ii), j represents type (iii) and k-l represent type (iv). The background in subgraphs a-h and j represent glacier-covered areas. Subgraph i shows several closed polylines, which does not fill background color. The different background colors in subgraphs k-l represent different areas of the glacier surface after the European allocation.

[Figure]

**Figure A6: The revised Figure 15.**

18. Figure 15: In the first two figures of the second row, where is the abnormal start point? In the fourth figure of the second row, why it is due to abnormal DEM?

**Response:** The gray polylines in Figure 15 represent glacier outlines. Intersection points of glacier outline and $A_l$ is the anomaly starting point in the first two figures of the second row (subgraphs e and f). We added these intersection points to the figure and the corresponding legend (See figure A6).

In the figure (subgraph h) in the second row and fourth column, $A_l$ is a straight line. There are generally two reasons for this situation: (i) the corresponding area is a flat surface with a slope of almost zero; (ii) the topography of the corresponding region is extremely complex, and the quality of DEM is too poor. The elevation values in a region are almost same because they are derived by interpolation. In the accumulation region of a glacier, the latter accounts for the vast majority. Thus, it is believed that this situation is caused by the abnormal DEM.

**Technical Corrections:**

Page 5, Line 85: Consider to change "arcpy" to "ArcPy".

**Response:** It has been modified.

Figure 2: In the part of the extraction of centerlines, it seems that $G_{cline}$ and $G_{gcf}$ should switch their position according to the author's definition.

**Response:** We rewrote acronym of each parameter to clarify their meanings, listed in the Appendix A (Table A1) and can also be found in the response to Comment 1.

Page 9, Line 148: In the third part of the post-processing, is it "numbers" or "members"?

**Response:** It should be "members", and refers to the elements that make up a line cluster.

Figure 9: Consider changing "inexact" to "inaccurate" for consistency.

**Response:** It has been modified.

Figure 12: Consider changing "$DL$" to "$D_L$" in the figure caption. For the figure on the right-hand side, the blue color should represent the number of $+D_L$, if I understand it correctly.

**Response:** It has been modified.

**References:**

Machguth, H., and Huss, M.: The length of the world's glaciers— a new approach for the global calculation of center lines, The Cryosphere, 8, 1741-1755, doi: 10.5194/tc-8-1741-2014, 2014.

Kienholz, C., Rich, J. L., Arendt, A. A., and Hock, R.: A new method for deriving glacier centerlines applied to glaciers in Alaska and northwest Canada, The Cryosphere, 8, 503-519, doi: 10.5194/tc-8-503-2014, 2014.

---

## Author Response (AR2)

**Responses to Editor Decision to manuscript TC-2020-294**

Dear Editor,

Thank you very much for your valuable comments to improve this manuscript. We responded point by point to each comment as listed below, along with a clear indication of the location of the revision.

If you have any queries, please don't hesitate to contact us at the address below. Looking forward to hearing from you.

Thank you and best regards.
Sincerely,

Dahong Zhang
Email: zhangdh_yx@163.com

**Please Notes:** Text in BLACK is the editor's comments and our responses are marked in BLUE. In addition, the notation used to locate the changes first defines the page number, then the line number(s). For example, **P4L15** means that the described modification to the manuscript can be found on the 15th line on the 4th page in the track-changes file.

**Comments:**

L20: were more superior => were superior

Response: It has been modified. **(P1L20)**

L20: provided => provides

Response: It has been modified. **(P1L20)**

L24: Glacier is => Glaciers are

Response: It has been modified. **(P2L24)**

L35: glacier centerline => the glacier centerline

Response: It has been modified. **(P2L35)**

L37: one-dimensional glacier model => one-dimensional glacier models

Response: It has been modified. **(P2L37)**

L103: was given => is given

Response: It has been modified. **(P5L103)**

L181: simple glacier => simple glaciers

Response: It has been modified. **(P11L181)**

L181: compound glacier => compound glaciers

Response: It has been modified. **(P11L181)**

In addition, we found and then changed two mistakes: (i) the wrong Eq.1 has been corrected **(P5L104)** and (ii) the default value of $P_9$ in Table 1 was not up-to-date and has been updated **(P6L108)**.